

# Simultaneous assimilation of ozone profiles from multiple UV-VIS satellite instruments

Jacob van Peet[1], Ronald van der A[1,2], Hennie Kelder[3], and Pieternel Levelt[1,4]

[1]Royal Netherlands Meteorological Institute (KNMI), De Bilt, The Netherlands
[2]Nanjing University of Information Science & Technology (NUIST), Nanjing, China
[3]Eindhoven University of Technology, Eindhoven, The Netherlands
[4]Delft University of Technology, Delft, The Netherlands

*Correspondence to:* van Peet (peet@knmi.nl), van der A (avander@knmi.nl)

**Abstract.** A three-dimensional global ozone distribution has been derived from assimilation of ozone profiles that were observed by satellite. By simultaneous assimilation of ozone profiles retrieved from the nadir looking satellite instruments Global Ozone Monitoring Experiment 2 (GOME-2) and Ozone Monitoring Instrument (OMI), which measure the atmosphere at different times of the day, the quality of the derived atmospheric ozone field has been improved. The assimilation is using an

extended Kalman filter in which chemical transport model TM5 has been used for the forecast. The combined assimilation of both GOME-2 and OMI improves upon the assimilation results of a single sensor. The new assimilation system has been demonstrated by processing 4 years of data from 2008 to 2011. Validation of the assimilation output by comparison with sondes show that biases vary between -5% and +10% between the surface and 100 hPa. The biases for the combined assimilation vary between -3% and +3% in the region between 100 and 10 hPa where GOME-2 and OMI are most sensitive. This is a strong

improvement compared to direct retrievals of ozone profiles from satellite observations.

## 1 Introduction

Depending on the altitude, ozone in the Earth's atmosphere has different effects. In the stratosphere, ozone filters the harmful ultraviolet part from the incoming solar radiation, preventing it from reaching the surface. Near to the surface, ozone is a pollutant, which has negative effects on human health and can reduce crop yields. At the same time, ozone is a greenhouse gas

with an important role in the temperature of the atmosphere.

Because of the important role ozone has in climate change, it has been designated an Essential Climate Variable (ECV) by the Global Climate Observing System (GCOS) of the World Meteorological Organisation (WMO) (WMO, 2010). In the GCOS report, it is stressed that the full three dimensional distribution of ozone is required.

The European Space Agency (ESA) has initiated the Climate Change Initiative (CCI) programme, which aims at long-

term time series of satellite observations of the ECVs (http://cci.esa.int/). One of the sub-programs is the Ozone CCI project (http://www.esa-ozone-cci.org/) that focuses on homogenized data sets of total ozone from different sensors (Lerot et al., 2014), stratospheric ozone distribution from limb and occultation observations (e.g. Sofieva et al., 2013) and the vertical ozone distribution from nadir observations (e.g. Miles et al., 2015). Long term ozone datasets were also produced by the European





Centre for Medium-Range Weather Forecasts (ECMWF) reanalyses such as ERA-40 (Uppala et al., 2005) and its successor ERA-Interim (Dee et al., 2011). Although primarily intended for improvement of the weather forecast, the assimilation of ozone is an integral part of theses reanalyses. It is described in more detail for ERA-40 in Dethof and Hólm (2004) and for ERA-Interim in Dragani (2011). Total ozone column measurements from different satellite instruments were assimilated into

a chemical transport model for the Multi Sensor Reanalysis (MSR) of ozone (van der A et al., 2010, 2015), spanning a 42 year period between 1970 and 2012.

Vertical ozone measurements from space-based Ultra Violet (UV) instruments started with the Solar Backscatter UltraViolet ((S)BUV) instruments from 1970 onwards on different satellites (e.g. Bhartia et al., 2013). Later, satellite instruments with higher resolution and increased spectral coverage were launched, for example Global Ozone Monitoring Experiment

(GOME) onboard ERS-2 in 1995 (Burrows et al., 1999), SCanning Imaging Absorption spectroMeter for Atmospheric CHartographY (SCIAMACHY) onboard Envisat in 2002 (Bovensmann et al., 1999), Ozone Monitoring Instrument (OMI) onboard Aura in 2004 (Levelt et al., 2006) and Global Ozone Monitoring Experiment 2 (GOME-2) onboard MetOpA/B in 2006/2012 (Callies et al., 2000; Munro et al., 2016). Each location on Earth is typically once or twice a day observed by these satellites, so it is not possible to get global coverage at a specific time of the day. The retrieved ozone profiles from Ultra Violet-VISible

(UV-VIS) nadir observations have a limited vertical resolution due to the smoothing effect in the retrieval (e.g. Rodgers, 1990). The vertical resolution varies between 7 and 15 km (see e.g. Hoogen et al., 1999). To derive gridded 3D ozone distributions at fixed time intervals we use data assimilation, which combines the information present in the model and the observations, giving the optimal estimate of the ozone concentration. Either the retrieved ozone data, or the radiance data from the instrument can be assimilated into the model. Migliorini (2012) showed that both methods are equivalent. However, assimilating retrieved

ozone data considerably simplifies the observation operator and reduces the number of measurements to assimilate. Since the measurement, averaging kernel and error covariance matrix are all used in our assimilation algorithm, all information gained from the retrieval is also present in the resulting assimilated model fields.

Two commonly used types of data assimilation are 4DVAR and (ensemble) Kalman filtering. For example, ozone profiles and total columns from different instruments (such as GOME) were assimilated using a 4DVAR assimilation scheme in the

production of the European Centre for Medium-Range Weather Forecasts (ECMWF) ERA-Interim reanalysis (see e.g. Dragani, 2011). The Belgian Assimilation System for Chemical ObsErvations (BASCOE, http://bascoe.oma.be/, Errera et al. (2008)) is a stratospheric 4DVAR data assimilation system for multiple chemical species including ozone and nitrogen dioxide. BASCOE is used in the MACC and CAMS projects for atmospheric services, the stratospheric ozone analyses from the MACC project are evaluated in Lefever et al. (2015). Recently, BASCOE has been coupled to the Integrated Forecast System of the ECMWF

(Huijnen et al., 2016). 4DVAR is well suited to assimilate large amounts of observations, and the analysis provides a smooth field at the time of the assimilation. However, there are two disadvantages of 4DVAR with respect to Kalman filter techniques. First, 4DVAR requires the development and maintenance of an adjoint model, which is a complicated process. Second, it does not produce an estimate of the uncertainty in the ozone field.

The model covariance matrix is an integral and essential part of a Kalman filter, but it is difficult to derive and computa-

tionally expensive in the analysis calculation. Therefore, most Kalman filter implementations try to approximate the model



covariance matrix. In the ensemble Kalman filter a selection of the ensemble members can be used to approximate the model covariance (see e.g. Evensen, 2003; Houtekamer and Zhang, 2016). Miyazaki et al. (2012) used an ensemble Kalman filter to assimilate different trace gas measurements from multiple satellite instruments into a chemical transport model.

In this research, we follow the Kalman filter approach described in Segers et al. (2005), where the model covariance matrix is parameterised into a time dependent standard deviation field and a time independent correlation field. The algorithm was updated and used by de Laat et al. (2009) to subtract the assimilated stratospheric ozone column from the total column in order to obtain a tropospheric ozone column. We have implemented several major updates and improvements in the algorithm compared to the version of de Laat et al.. We check the observational error characterisation, redefine the model error growth and derive a new correlation matrix for the ozone field. The new algorithm is the first that simultaneously assimilates nadir
ozone profiles from multiple high spectral resolution satellite instruments. We demonstrate the new algorithm by assimilating ozone profile observations from GOME-2 and OMI for the period 2008-2011 into the chemical transport model TM5 (e.g. Krol et al., 2005). To minimise the bias between the two instruments, we developed a bias correction based on ozone sondes to be applied to the observations before assimilation. A bias correction based on total column measurements from ground stations was earlier used for the Multi Sensor Reanalysis of total ozone (van der A et al., 2015). Since we assimilate ozone profiles we
require an altitude dependent bias correction for which ozone soundings are selected.

In section 2 we briefly describe the ozone profile observations, and in section 3 the chemical transport model is described that we use for the data assimilation. Section 4 gives a short overview of the assimilation algorithm, section 5 describes the improvements applied to the assimilation algorithm and the results will be shown in section 6. In section 7 we demonstrate the performance of the assimilation algorithm over the Tibetan Plateau. A discussion of the results is given in section 8 and the
conclusions are presented in section 9.

## 2 Observations

Data from the UV-VIS satellite instruments GOME-2 and OMI are available for the last 10 years. GOME-2 (Callies et al., 2000; Munro et al., 2016) was launched on-board METOP-A in 2006. The instrument measures the solar light reflected by the Earth's atmosphere between 250 and 790 nm. For the retrievals used in this research, the radiance measurements are binned in
the cross track and along track directions such that the ground pixels measure approximately 160×160 km. The ozone profiles for GOME-2 are retrieved with the OPERA retrieval algorithm, which is described in (van Peet et al., 2014). We increased the number of layers in this study from 16 to 32 for more accurate radiative transfer model results.

OMI (Levelt et al., 2006) was launched on-board Aura in 2004. The instrument measures the solar light reflected by Earth's atmosphere between 270 and 500 nm. One important difference between OMI and GOME-2 is that OMI does not use a
scanning mirror like GOME-2, but a fixed 2D CCD detector. One dimension of the detector is used to cover the spectral range, the other is used to cover the cross-track direction. The ground pixels for the profiles retrieved from the UV-VIS spectrum measure approximately 65×48 km, with the size increasing towards the edge of the swath. OMI has two UV-VIS channels that are used in ozone profile retrieval: UV1 and UV2. UV1 has thirty cross-track pixels, while UV2 has sixty cross-track pixels.



The UV2 pixels are therefore averaged to coincide with the UV1 pixels. A description of the OMI ozone retrieval algorithm and validation results with respect to ground measurements and other satellite instruments can be found in Kroon et al. (2011).

The algorithms used to retrieve the ozone profiles from GOME-2 and OMI are both based on an optimal estimation technique. The state of the atmosphere is given by the state vector $x$, while the measurement is given by the measurement vector $y$ and error $\epsilon$. These two vectors are related by the forward model $\mathbf{F}$ according to $y = \mathbf{F}(x) + \epsilon$. Following the maximum a-posteriori approach (Rodgers, 2000), the solution is given by:

$$\hat{x} = x_a + \mathbf{A}(x_t - x_a) + \mathbf{G}\epsilon \tag{1}$$

$$\hat{\mathbf{S}} = (\mathbf{I} - \mathbf{A})\mathbf{S}_a \tag{2}$$

$$\mathbf{A} = \mathbf{GK} = \mathbf{S}_a\mathbf{K}^T\left(\mathbf{KS}_a\mathbf{K}^T + \mathbf{S}_\epsilon\right)^{-1}\mathbf{K} \tag{3}$$

where $\hat{x}$ is the retrieved state vector, $x_a$ is the a priori, $\mathbf{A}$ is the averaging kernel, $x_t$ is the "true" state of the atmosphere, $\mathbf{G}$ is the gain matrix ($\mathbf{S}_a\mathbf{K}^T\left(\mathbf{KS}_a\mathbf{K}^T + \mathbf{S}_\epsilon\right)^{-1}$), $\mathbf{G}\epsilon$ the retrieval noise, $\hat{\mathbf{S}}$ is the retrieved covariance matrix, $\mathbf{I}$ is the identity matrix, $\mathbf{S}_a$ is the a priori covariance matrix, $\mathbf{K}$ is the weighting function matrix or Jacobian (it gives the sensitivity of the forward model to the state vector) and $\mathbf{S}_\epsilon$ is the measurement covariance matrix.

The averaging kernel can also be written as $\mathbf{A} = \partial\hat{x}/\partial x_t$ and gives the sensitivity of the retrieval to the true state of the atmosphere. The trace of $\mathbf{A}$ gives the degrees of freedom for the signal (DFS). When the DFS is high, the retrieval has learned more from the measurement than in the case of a low DFS, when most of the information in the retrieval will depend on the a priori. The total DFS can be regarded as the total number of independent pieces of information in the retrieved profile. The rows of $\mathbf{A}$ indicate how the true profile is smoothed out over the layers in the retrieval and are therefore also called smoothing functions. Ideally, the smoothing functions peak at the corresponding level and the half-width is a measure for the vertical resolution of the retrieval.

Because the sensitivity of the retrieval to the vertical ozone distribution is represented by the averaging kernel, it is important to include the averaging kernel in the assimilation algorithm. Together the retrieved state vector, averaging kernel and error covariance matrix represent all information gained from the retrieval (Migliorini, 2012).

## 3 Chemical transport model TM5

The model used in the assimilation is a global chemistry transport model called TM5 (Tracer Model, version 5), see Krol et al. (2005) for an extended description. The (tropospheric) chemistry of TM5 has been evaluated in Huijnen et al. (2010) and included into the integrated forecasting system of the ECMWF (Flemming et al., 2015).

In the current model setup used for the assimilation of the ozone profiles, TM5 runs globally with grid cells of 3° longitude × 2° latitude, on 45 pressure levels. The pressure levels are a subset of the 91 level pressure grid from the European Centre for Medium-Range Weather Forecasts (ECMWF). The meteo data used to drive the TM5 tracer transport are taken from the ECMWF operational analysis fields, produced on these 91 pressure levels.



Above 230 hPa, ozone chemistry is parameterised according to the equations described by Cariolle and Teyssèdre (2007), using the parameters of version 2.1. In the troposphere, the ozone concentrations are nudged towards the Fortuin & Kelder climatology (Fortuin and Kelder, 1998), with a relaxation time that increases from 0 days at 230 hPa to 14 days at 500 hPa and lower. No other trace gasses are modelled in this setup, which makes this version of TM5 fast and computationally cheap.

5 Ozone concentrations are prevented from following the model equilibrium state too closely by the frequent confrontation of the model with the observations during the assimilation process.

## 4 Assimilation algorithm

The assimilation algorithm uses a Kalman filter, and is described in Segers et al. (2005). The state vector $\boldsymbol{x}$ and the measurement vector $\boldsymbol{y}$ are given by:

$$\boldsymbol{x}_{i+1} = M\left(\boldsymbol{x}_i\right) + \boldsymbol{w}_i, \quad \boldsymbol{w}_i \sim N\left(\boldsymbol{0}, \mathbf{Q}_i\right) \tag{4}$$

$$\boldsymbol{y}_i = H\left(\boldsymbol{x}_i\right) + \boldsymbol{v}_i, \quad \boldsymbol{v}_i \sim N\left(\boldsymbol{0}, \mathbf{R}_i\right) \tag{5}$$

where $M$ is the model that propagates the state vector in time. It has an associated uncertainty $\boldsymbol{w}$, which is assumed to be normally distributed with zero mean and covariance matrix $\mathbf{Q}$. The observation operator $H$, which includes the averaging kernel, gives the relation between $\boldsymbol{x}$ and $\boldsymbol{y}$. The uncertainty in $\boldsymbol{y}$ is given by $\boldsymbol{v}$, which is also assumed to have zero mean and

15 covariance matrix $\mathbf{R}$ (which is identical to $\hat{\mathbf{S}}$ in the retrieval equations). In matrix notation, the propagation of the state vector and its covariance matrix ($\mathbf{P}$) are given by:

$$\boldsymbol{x}_{i+1}^f = M\left(\boldsymbol{x}_i^a\right) \tag{6}$$

$$\mathbf{P}_{i+1}^f = \mathbf{M}\mathbf{P}_i^a\mathbf{M}^T + \mathbf{Q}_i \tag{7}$$

where $\boldsymbol{x}^a$ is the state vector at time $t = i$, after assimilation of the observations. The observations are assimilated according to:

$$\boldsymbol{x}_i^a = \boldsymbol{x}_i^f + \mathbf{K}_i\left(\boldsymbol{y}_i - H\left(\boldsymbol{x}_i^f\right)\right) \tag{8}$$

$$\mathbf{P}_i^a = \left(\mathbf{I} - \mathbf{K}_i\mathbf{H}_i\right)\mathbf{P}_i^f \tag{9}$$

$$\mathbf{K}_i = \mathbf{P}_i^f\mathbf{H}_i^T\left(\mathbf{H}_i\mathbf{P}_i^f\mathbf{H}_i^T + \mathbf{R}_i\right)^{-1} \tag{10}$$

where $\mathbf{K}$ is called the Kalman gain matrix and $H$ the observation operator. The matrix $\mathbf{H}$ is the sensitivity of the observation operator with respect to the state.

25 The observation operator interpolates the model field to the observation location, converts the model units to the retrieval units and takes the smoothing of the satellite instruments into account by incorporating the averaging kernel. The model grid cells are $3 \times 2°$ (longitude $\times$ latitude), much larger than the satellite ground pixels and therefore no horizontal interpolation is needed. The model profile, expressed DU/layer, is converted to the pressure levels of the retrieval grid by applying a simple linear interpolation in the $^{10}\log(\text{hPa})$ domain. Finally, the observation operator $H$ is formed by applying the averaging kernel



**A** to the difference between the state vector $\boldsymbol{x}$ and the a-priori profile $\boldsymbol{y}_a$ used in the retrieval:

$$H\left(\boldsymbol{x}\right) = \mathbf{A}\left(\mathbf{BC}\boldsymbol{x} - \boldsymbol{y}_a\right) \tag{11}$$

with **C** the unit conversion (from the models $\mathrm{kg/gridcell}$ to the observations $\mathrm{DU/layer}$), **B** the vertical interpolation. The sensitivity matrix **H** used in equations (9) and (10) is constructed as $\mathbf{H} = \mathbf{ABC}$.

In general, the number of elements in an ozone profile is much larger than the degrees of freedom (about 5 to 6). We can therefore reduce the number of data points per profile by taking the singular value decomposition of the **A**, and only retain the vectors with a singular value larger than 0.1. The profiles and matrices are transformed accordingly.

The computational cost of the assimilation algorithm can be further reduced by minimising the size of the model covariance matrix **P**. The TM5 model runs in the current setup on a horizontal grid of $2° \times 3°$ (latitude $\times$ longitude) on 44 layers, which

makes the size of the covariance matrix $(475200)^2$ elements. A number of different approaches exist to minimise the size of the model covariance matrix. For example, in Eskes et al. (2003), the number of dimensions is reduced by only assimilating total columns, while the horizontal correlation depended only on the distance between the model grid cells. Here, we follow the approach described by Segers et al. (2005), by parameterising the model covariance into a time dependent standard deviation field and a constant correlation field. Each time step, the model's advection operator is applied to the standard deviation field.

The error growth (i.e. the addition of **Q** in equation (7) is modelled by a simple mathematical function, more details can be found in section 5.2. The model covariance matrix can now be calculated according to:

$$\mathbf{P} = \mathfrak{D}\left(\boldsymbol{\sigma}\right)\mathbf{C}\mathfrak{D}\left(\boldsymbol{\sigma}\right) \tag{12}$$

with $\mathfrak{D}\left(\boldsymbol{\sigma}\right)$ a matrix with the values of the standard deviation $\boldsymbol{\sigma}$ on the diagonal and **C** the correlation matrix. The correlation matrix is calculated differently than in Segers et al. (2005), more details can be found in section 5.3.

Unfortunately, the $\left(\mathbf{H}_i\mathbf{P}_i^f\mathbf{H}_i^T + \mathbf{R}_i\right)$ matrix in the Kalman filter (equation (10)) is badly conditioned, which makes the inversion sensitive to noise. Therefore, the eigenvalue decomposition of this matrix is calculated and the measurements are projected on the largest eigenvalues, which in total represent 98% of the original trace of the matrix.

For the numerical stability of the assimilation algorithm, the difference between the observation and the model should not be too large. A filter is implemented that rejects the observation when the absolute difference between the observation and the

model forecast is larger than three times the square root of the sum of the variance of the observation and the variance of the forecast:

$$\mathrm{abs}\left(\boldsymbol{y}_i - H\left(\boldsymbol{x}_i^f\right)\right) \geq 3\sqrt{\sigma_{y_i}^2 + \sigma_{x_i^f}^2} \tag{13}$$

with $\sigma_{\boldsymbol{y}}$ and $\sigma_{\boldsymbol{x}}^f$ the standard deviation of the observation $\boldsymbol{y}$ and the forecast $H(\boldsymbol{x}^f)$ for layer $i$ respectively. Note that this is done on a layer-by-layer basis, i.e. if in one layer the difference is too large, the whole observed profile is discarded.

Not all available ozone profiles can be assimilated into TM5 because the computational cost would be too high. Therefore 1 out of 3 GOME-2 profiles and 1 out of 31 OMI profiles are used. These numbers are chosen such that more or less the same number of observations are assimilated for each instrument, taking into account the decrease in available pixels due to the row





anomaly in OMI. For OMI, the outermost pixels on each side of the swath are neglected, because of the large area of these pixels. Of the resulting retrievals, only cloud free scenes (cloud fraction $\leq 0.2$) are assimilated in order to get the maximum amount of information from the troposphere.

## 5 Improvements of the assimilation algorithm

The first version of our assimilation algorithm was described in Segers et al. (2005). They assimilated GOME ozone profiles for the year 2000. This dataset was extended to the period 1996–2001 by de Laat et al. (2009) who derived tropospheric ozone for this period. The assimilated GOME observations in the previous algorithm version had a pixel size of $960 \times 100$ km, much larger than the pixels in the current research. Since 2009, the assimilation algorithm has been further developed and improved for use with GOME-2. The improved resolution of GOME-2 and OMI ozone profiles and improved retrievals offer
new possibilities, but also require adaptations in the data assimilation. It is the first time that ozone profiles from two nadir looking instruments, GOME-2 and OMI, are assimilated simultaneously. This has resulted in a significant number of updates and improvements to the assimilation algorithm compared to the version described in Segers et al. and de Laat et al. (2009), which are outlined in the following sections.

### 5.1 Observational error characterisation

The covariance matrix of the observations that is used in the assimilation is composed of two components, the error on the spectral observations and the error of the a-priori information. Since the spectral errors affect the assimilation results, they are first verified, using the following method.

For a given wavelength two adjacent detector pixels may have a radiance or reflectance difference that depends on the slope of the spectrum. Given enough samples, the standard deviation of the mean difference is a good indication of the noise at that
particular wavelength. The relative difference $D$ is calculated as:

$$D = \frac{F(\lambda_1) - F(\lambda_2)}{0.5(F(\lambda_1) + F(\lambda_2))} \tag{14}$$

where $F$ is the radiance and $\lambda_1$ the wavelength in detector pixel 1 and $\lambda_2$ the wavelength in the adjacent pixel. Because the standard deviation is sensitive to outliers, a Gaussian distribution is fitted to the data. The fitted standard deviation is multiplied with the spectrum and compared to the reported noise in the level-1 data.

For GOME-2, we checked four days in 2008: 15 March, 25 June, 26 September and 25 December. On December 10th, 2008 the band 1A/B boundary was shifted from approximately 307 nm to 283 nm and the integration time for band 1B decreased from 1.5s to 0.1875s in this wavelength range. Therefore, the data for the first three days are combined, while the data for 25 December is treated separately. The analysis was performed for different subsets, such as latitude, solar zenith angle and viewing angle, but results are shown for latitude only.





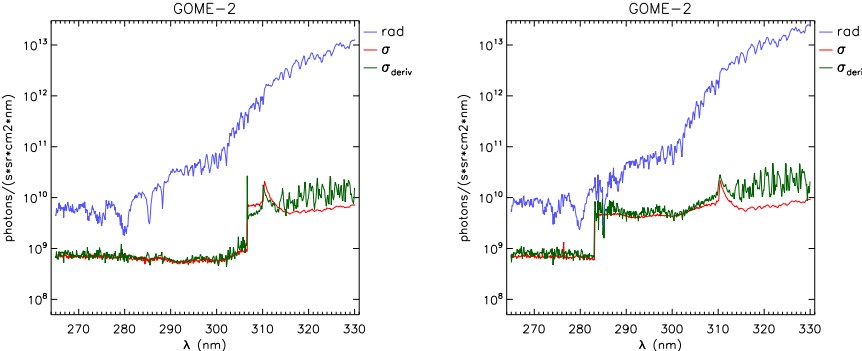

**Figure 1.** GOME-2 METOP-A radiance spectra calculated by OPERA: before (left) and after (right) the wavelength shift from 307 to 283 nm. The blue and red lines are the radiance and uncertainty that are used in OPERA. The green line shows the fitted standard deviations of the relative difference (see equation (14)) multiplied by the radiance.

Figure 1 shows the resulting GOME-2 radiance spectra for all wavelengths. It should be noted that the these results are made using spectral data derived with the GOME Data Processor (GDP) version 5.3. The older version GDP 4 uses a different noise model, which yielded too large errors.

The wavelength grid for OMI varies with the location across the detector, so the error verification has been performed with a dependence on the cross-track position. An example radiance spectrum along with the uncertainties is shown in the left panel of Figure 2. There is a jump in the spectral uncertainty (the red line) around 300 nm, which might be related to a change in the gain settings for the detector. For the selected pixel, the gain changes with a factor of 10 at 300nm.

On February 1, 2010, a L0 to L1b processor update was implemented for OMI. The new processor version includes more detailed information on the row anomaly and a new noise calculation for the three channels UV1, UV2 and VIS. More information can be found on the following website: http://projects.knmi.nl/omi/research/calibration/GDPS-History/GDPS_V113.html The new noise calculation was investigated by taking the radiance differences determined a few days after the update. The resulting radiance spectra are given in the right panel of Figure 2. The uncertainties in the L1 observations after the L0 to L1b processor update are about a factor of 5 smaller than the uncertainties derived according to the method described above.

In general, the spectral uncertainties for GOME-2 show a good agreement with our fitted uncertainties and therefore we simply use the uncertainties provided with the observations. The spectral uncertainties for OMI show a good agreement with our fitted uncertainties before the processor update, but are too small afterwards. The consequences of these smaller uncertainties will be shown in section 6. Since we use the OMI observations as they are, we are not able to correct for the spectral uncertainties in the retrieval.

### 5.2 Model Error Growth

In section 4 we explained that using the full covariance propagation from the Kalman filter equations is too computational intensive, instead we parameterise the model covariance matrix into a time dependent standard deviation field and a time





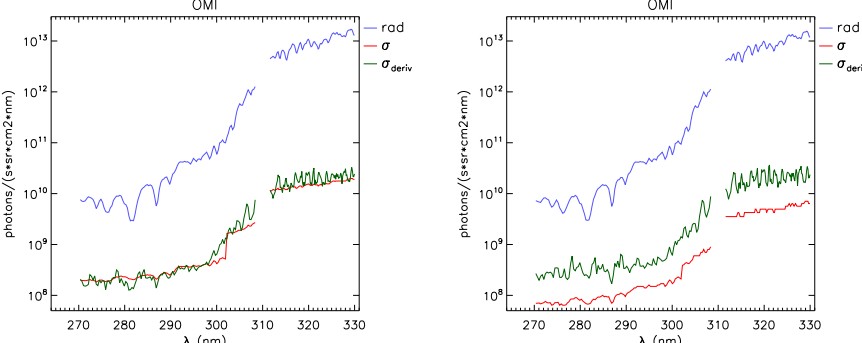

**Figure 2.** OMI radiance spectrum used in the retrieval, the area around 310 nm is not used. The blue and red lines are the radiance and uncertainty respectively. The green line shows the fitted standard deviations of the relative difference (see equation (14)) multiplied by the radiance. Left plot before the L0 to L1b processor update: date = 25-02-2006, lon = $145.2°$, lat = $-20.3°$; right plot after the update: date = 5-2-2010, lon = $138.0°$, lat = $-28.0°$.

independent correlation field. The advection operator is applied to the standard deviation field, and the model error growth is modelled by applying a simple empirical relation.

In the previous version of the assimilation algorithm, the error growth for the total column was modelled by the function $e(t) = At^{1/3}$ (Eskes et al., 2003). The error for the total column was distributed over the layers in the profile, proportional to

the partial columns in each layer (Segers et al., 2005). Deriving a similar relation on a layer-by-layer basis was not successful, because the error can grow unlimited using this error growth description. Especially during the polar night this might lead to unrealistic high error values.

Therefore, we use the following function

$$e(t) = \frac{at}{b+t} \tag{15}$$

where $a$ and $b$ are parameters which can be determined by fitting the observation minus forecast RMS as a function of time (see Eskes et al. (2003), figure 2). The parameter a is the maximum relative error of the model at a particular altitude. At $t = b$, the error is $0.5a$, therefore $b$ is a measure of how fast the error grows after a measurement has been assimilated. The best results are obtained using $b = 2$ (days) and let the value of a vary over altitude. The values of a are determined by comparing the free model run (i.e. no assimilation) with all sondes for 2008. These results include a representation error due to the grid cell size

of the model, and is therefore an overestimation of the real model error. Therefore, all collocations that are more than $3\sigma$ from the mean are discarded. The RMS values of the resulting collocations are used as values for a (see Table 1). For the error of the layers above the maximum altitude of the sondes (about 5 hPa), a has been set to the same value as the last layer below the maximum altitude.





**Table 1.** Values of $a$ as a fraction of the partial column at different altitudes.

| hPa | $a$ | hPa | $a$ | hPa | $a$ |
|---|---|---|---|---|---|
| surface | 0.22 | 131.50 | 0.44 | 19.38 | 0.13 |
| 678.57 | 0.22 | 100.03 | 0.50 | 14.74 | 0.15 |
| 516.19 | 0.20 | 76.09 | 0.43 | 11.21 | 0.18 |
| 392.68 | 0.28 | 57.88 | 0.21 | 8.53 | 0.21 |
| 298.72 | 0.41 | 44.03 | 0.18 | 6.49 | 0.25 |
| 227.24 | 0.43 | 33.50 | 0.13 | 4.94 | 0.34 |
| 172.86 | 0.34 | 25.48 | 0.12 | TOA | 0.34 |

## 5.3 Model correlation matrix

In order to calculate the time independent correlation field, we follow the National Meteorological Center's method (NMC-method) to determine the correlation in the model (see Parrish and Derber, 1992; Segers et al., 2005). Segers et al. (2005) used a reference run based on 6-hourly meteorological forecasts as the starting point for forecast runs that last 9 days and start at 12 UTC. After a spin-up period, 9 forecast fields per day are available which can be used to determine the correlation in ozone. Differences between the ozone concentration in these runs are due to the different meteorological inputs. Since the overpass frequency of GOME is three days, the forecast field from the run started 3 days before the current date was used to derive the correlations in the ozone field. This choice also best matched the correlation length found by Eskes et al. (2003), where total columns were assimilated instead of profiles.

We use a slightly different approach as Segers et al. (2005) because their method neglects uncertainties due to the chemistry parameterisation. Also, the forecast lag of three days is not compatible with GOME-2 and OMI, which have daily global coverage. Our reference run is the result of the assimilation of profile observations for April 2008, which we consider the true state of the atmosphere. Using the analysis field at 0 UTC, a model run without assimilation (a free model run) is started for a duration of 10 days. After the first 10 days, there are 11 model fields for a given date at 12 UTC: 1 from the assimilation run and 10 from the free model runs (see Figure 3).

The difference between the assimilation and free model runs is used to determine the correlations between all pairs of grid cells in the vertical direction (constant location), in the East-West direction (constant latitude and altitude), and in the North-South direction (constant longitude and altitude). The correlations are determined as a function of the distance. Since the East-West distance between two grid cells is larger at the equator than near the poles, the East-West correlation also depends on the latitude. The calculated correlations as function of distance are fitted with a Gaussian distribution (with correlations less than 0.01 set to zero). Both the calculated and fitted correlations are shown in Figure 4. The fitted correlations are used in subsequent model runs as the time independent correlation field.




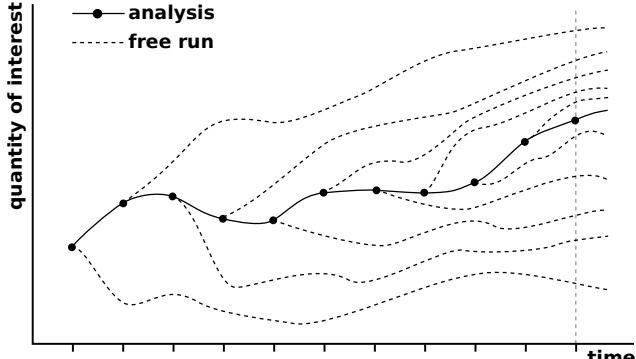

**Figure 3.** Determination of the TM5 correlation field. The solid line is an assimilation model run, the dashed lines are 10-day free model runs. After 10 days, there are 11 ozone fields for each given day which can be used to determine the correlations.

## 5.4 Ozone profile error characterisation and bias correction

The biases between two instruments should be as small as possible for a stable assimilation. Therefore, a bias correction as function of Solar Zenith Angle (SZA), Viewing Angle (VA) and time has been developed based on the results of the comparison with sondes. The bias correction factor is one minus the median of the relative deviation based on all collocated data in a given

year. All observations in a given year are multiplied by this correction factor.

Figure 5 shows the validation results for the four years of the assimilation period (2008–2011) of the GOME-2 and OMI profiles with ozone sondes downloaded from the World Ozone and Ultraviolet Radiation Data Center (WOUDC, WMO/GAW, 2016) for cloud free (cloud fraction $< 0.2$) retrievals. The sonde profiles have been interpolated on the pressure grid of the retrievals and convolved with the averaging kernels in order to take the vertical sensitivity of the satellite instruments into

account.

The bias of GOME-2 with respect to sondes varies between -1.1 and +1.7 DU (-7 and +7%) between 100 and 10 hPa, while for altitudes below 100 hPa the bias is about -0.3 DU (-4%). The bias of OMI varies between -4.5 and +2 DU (-8 and +15%) between 100 and 10 hPa, while below 10 hPa the bias is positive with a maximum value of 4 DU (+27%). The absolute biases cannot be compared directly because the layers of GOME-2 and OMI do not have the same thickness. Note that the remaining

biases for the top layers in Figure 5 are not exactly zero for the corrected observations, because the figure is drawn for latitude bands, while the bias correction is made using SZA and VA bins and the number of sondes used in the comparison at that altitude is much smaller than at lower altitudes. The number of sondes in the lowest and top layers is 1083 and 10 respectively for GOME-2 and 776 and 33 respectively for OMI.

## 6   Results and validation

We have assimilated GOME-2 (on Metop-A) and OMI ozone profiles for a period of 4 years between 2008-2011 using the Kalman filter algorithm described in the previous sections. In total, four model runs were performed: a 'free' model run





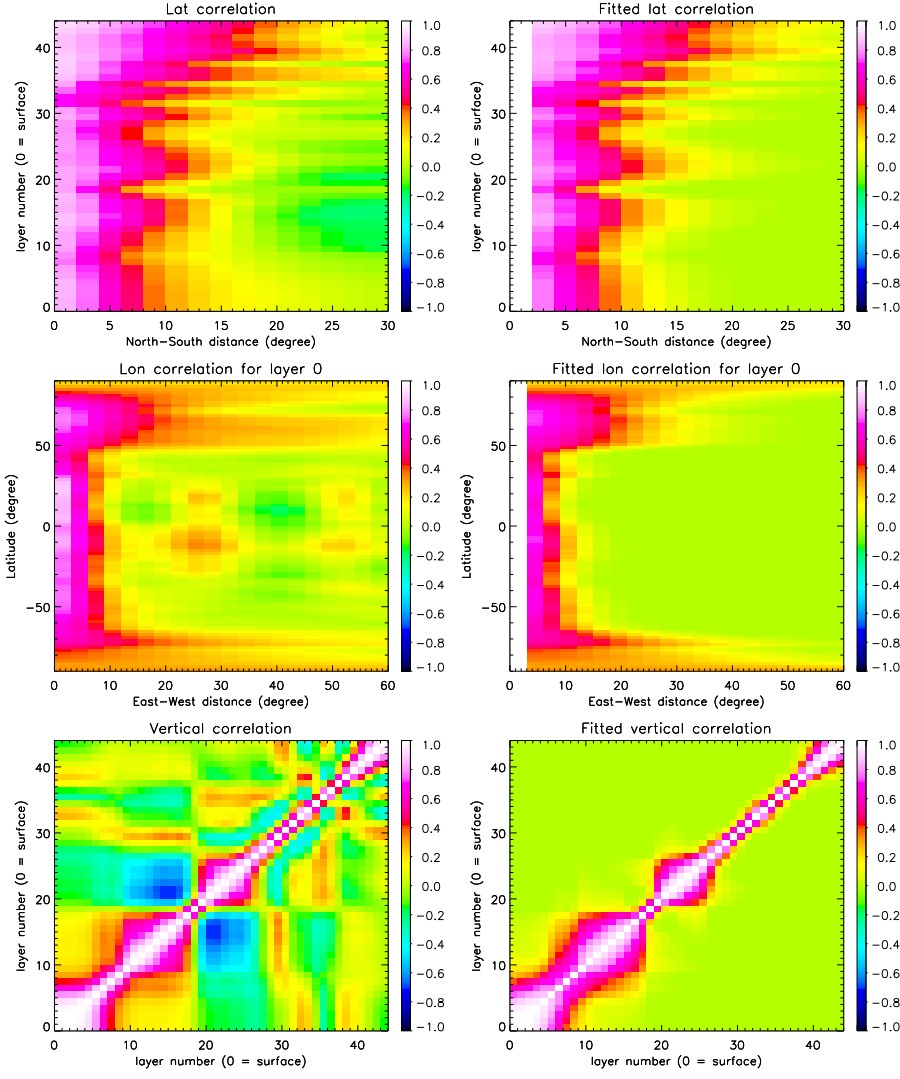

**Figure 4.** Calculated (left) and fitted (right) correlations for the latitudinal (top), surface layer longitudinal (middle) and vertical (bottom) directions.

without assimilation, a model run with assimilation of GOME-2 ozone profiles only, a model run with assimilation of OMI ozone profiles only and a model run with simultaneous assimilation of GOME-2 and OMI ozone profiles.



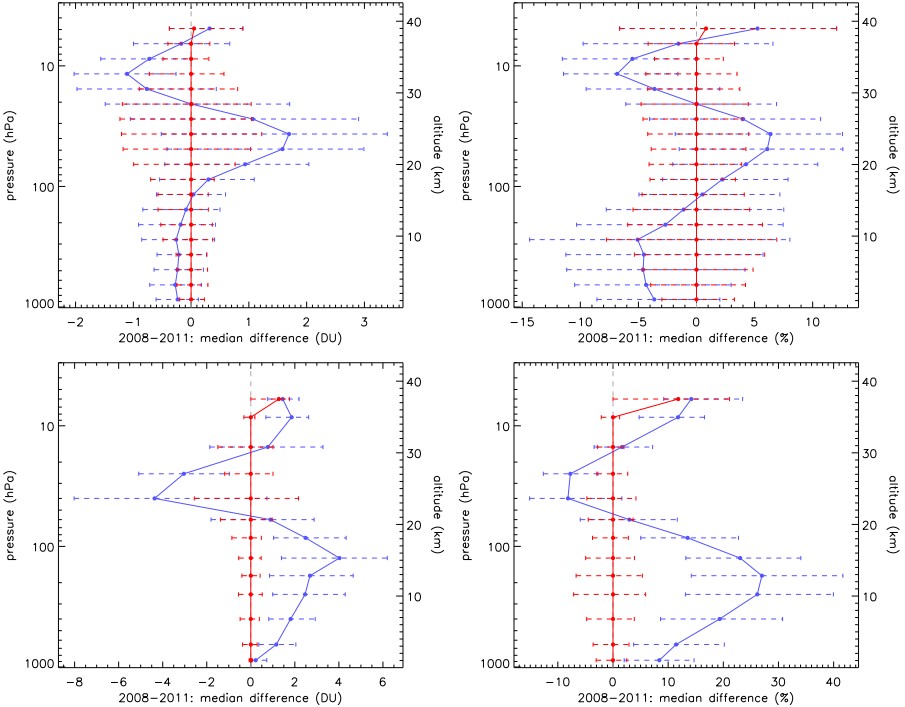

**Figure 5.** Validation results for 2008-2011 for GOME-2 at the top and OMI at the bottom. The left column shows the median absolute differences, the right column shows the median relative differences. The blue line indicates the original observations, the red line the bias corrected observations that have been used as input for the assimilation. The error bars indicate the range between the 25% and 75% percentiles. Note that the x-axis scale is different for each plot.

### 6.1 Altitude dependent OmF and OmA statistics

An important diagnostic of any assimilation system are the differences between the observations and the model (also known as innovations). In the following, we define the relative observation minus forecast (OmF) for layer $i$ as:

$$\mathrm{OmF}_i = \frac{\boldsymbol{y}_i - H\left(\boldsymbol{x}_i^f\right)}{0.5\left(\boldsymbol{y}_i + H\left(\boldsymbol{x}_i^f\right)\right)} \tag{16}$$

5   with $i$ the layer index, $\boldsymbol{y}$ the observed ozone profile, $H$ the observation operator and $\boldsymbol{x}^f$ the forecast profile of the model (see section 4). The layers in the retrievals of GOME-2 and OMI have a different thickness, which makes the comparison of the OmF between the two instruments not straightforward. Therefore, both $\boldsymbol{y}$ and $H\left(\boldsymbol{x}^f\right)$ have been regridded to the same pressure levels before calculating the OmF. This new pressure grid is defined by levels at 0, 6 and 12 km followed by levels every 2 km up to 60 km, corresponding to surface pressure up to 0.28 hPa. The observation minus analysis (OmA) is defined in a similar

10  way, but with $\boldsymbol{x}^f$ replaced with the analysis profile $\boldsymbol{x}^a$. Since the analysis field is a weighted average of the forecast model field and the observations, the OmA should be smaller than the OmF.





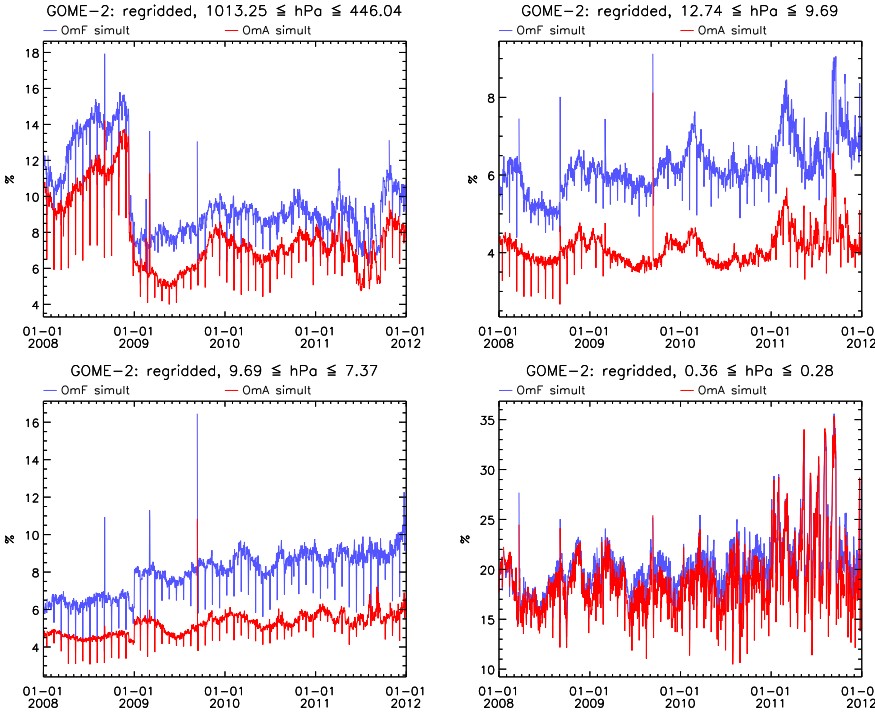

**Figure 6.** GOME-2 OmF (blue) and OmA (red) for the surface layer (top left), around 10 hPa (top right and bottom left) and around 0.3 hPa (bottom right). The OmF and OmA have been calculated for the regridded layers from the model run with simultaneous assimilation of GOME-2 and OMI.

In Figure 6, the GOME-2 OmF and OmA from the model run with simultaneous assimilation of GOME-2 and OMI for four different layers have been plotted. The ozone sondes that were used in deriving the bias correction and the validation of the results were required to have reached at least 10 hPa. Therefore the selected layers in Figure 6 are the surface layer, the layer just below and above 10 hPa, and the top layer of the new pressure grid around 60 km (0.3 hPa). In Figure 7, the OmF and OmA for the same layers have been plotted for OMI. In the first year of the assimilation period, the surface layer OmF and OmA for GOME-2 are higher than those for OMI. At the end of 2008, after the wavelength shift between GOME-2 band 1A/1B, the situation is reversed and the OmF and OmA for GOME-2 are lower than those for OMI. The OMI data show a more pronounced yearly cycle than GOME-2. After the beginning of 2010, the OmF and OmA for both instruments are very similar for the summer months June, July and August, but the winter time values for OMI are higher. For the layer just below the 10 hPa, the OmF and OmA for GOME-2 are about 1 percent point higher than for OMI. For the layer just above the 10 hPa, the OmF and OmA for GOME-2 start out lower than for OMI, but at the end of the assimilation period, the values are comparable. For the top layer, the OmF and OmA for GOME-2 are about 5 percent point higher than for OMI . In general, the OmF is about 2–4 percent point higher than the OmA, except for the top layer. There, the difference is in the order of 1 percent point, but the values vary much more than lower in the atmosphere.





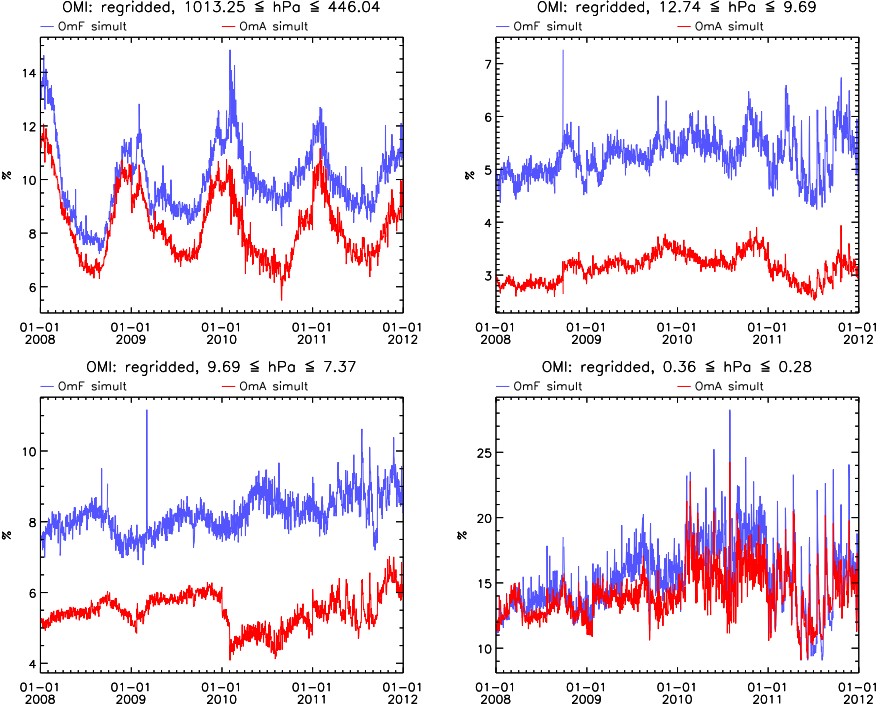

**Figure 7.** OMI OmF (blue) and OmA (red) for the surface layer (top left), around 10 hPa (top right and bottom left) and around 0.3 hPa (bottom right). The OmF and OmA have been calculated for the regridded layers from the model run with simultaneous assimilation of GOME-2 and OMI.

Both OmF and OmA for the GOME-2 assimilation run show regular decreases with a period of about one month. These decreases are caused by GOME-2 being operated in 'narrow-swath mode', when the swath is 320 km wide instead of 1920 km. For these narrow-swath observations, the model is closer to the retrieved profiles, causing a lower OmF/OmA. OMI also has a spatial zoom-in mode, which is activated about once a month, but these pixels are filtered out because they are too much

5 influenced by the row-anomaly and because the mapping between the UV-1 and UV-2 pixels changes with respect to the normal mode. Peaks in the OmF and OmA for the GOME-2 assimilation, such as after an instrument test period between 7 and 12 September 2009, can be related to periods of missing data.

Sudden changes in the OmF and OmA are visible for some altitudes for both instruments at the start of some years. One example is in the layer just above the 10 hPa for GOME-2 at the start of 2009 or at the start of 2010 for OMI. Since there

10 are no known instrumental or meteorological changes, the most likely cause is therefore the bias correction scheme for the observations, which changes its correction parameters at the start of each year.

Closer inspection of the OMI OmF and OmA change at the start of 2010 (see the lower left panel of Figure 7, shows that it actually consists of two steps: the first one at the start of the year, and the second one a month later. That second step is also present in the lower right panel (the layer around 0.3 hPa), where the change is about 5 percent point, but it is less clear due to





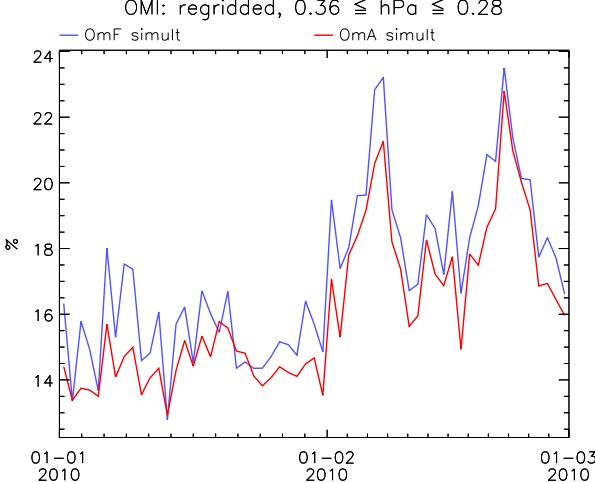

**Figure 8.** OMI OmF (blue) and OmA (red) for the layer around 0.3 hPa, zoomed in to a month before and after the L0 to L1b processor update. The OmF and OmA have been calculated for the regridded layers from the model run with simultaneous assimilation of GOME-2 and OMI.

the higher variability in the signal. Figure 8 shows the same data, but focused on the first two months of 2010. Both OmF and OmA increase by about 5 percent point from one day to the next. The increase is even larger (and more clearly visible) in the data from the single instrument assimilation run for OMI.

Comparison of Figure 6 and Figure 7 shows that the OmF and OmA for one instrument might be larger than for the other,

depending on the altitude. Which of the two instruments has a larger OmF or OmA value might also change over time. In other words, GOME-2 and OMI have a different sensitivity for different altitudes as represented by the averaging kernels. Assimilating the observations from these instruments simultaneously, increases the overall sensitivity of the assimilation.

Lower uncertainties in the spectra lead to lower uncertainties on the observations, which on its turn changes the balance between model and observations in the Kalman filter and affects the innovations. Because the variance of the observation is

lower, more pixels will be rejected by the OmF filter (see section 4 and Figure 9). The figure shows the number of assimilated observations for both GOME-2 and OMI from the single and simultaneous instrument assimilation. In the single instrument assimilation runs, the model error is adapted to the new situation after the processor update and the total number of assimilated observations does not change. For the simultaneous assimilation, the OmF differences become so large with respect to the uncertainties, that the OmF filter rejects observations from both GOME-2 and OMI, even though only the OMI data have

changed.

## 6.2 Altitude independent OmF and OmA statistics

In order to show the geographical distribution of the OmF and OmA, the absolute values for each layer were quadratically added and the square root was taken from the result. These column integrated OmF and OmA values were averaged on a daily



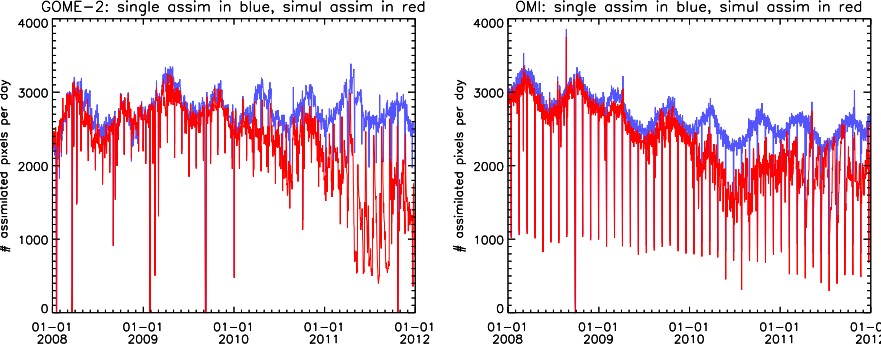

**Figure 9.** Number of assimilated observations from GOME-2 (left) and OMI (right). The blue lines represent the single instrument assimilation, the red lines the simultaneous assimilation.

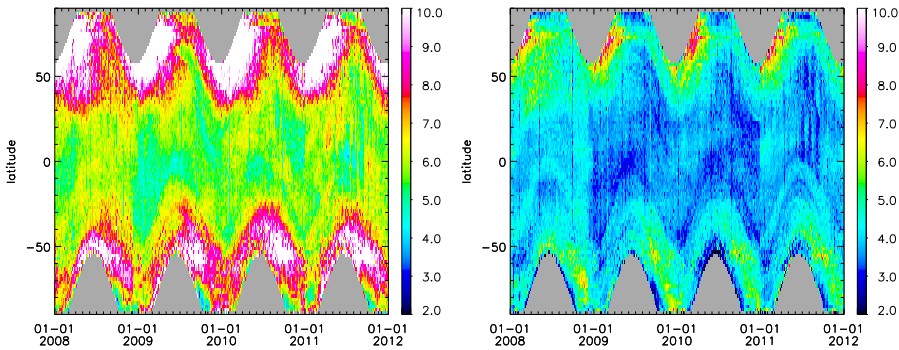

**Figure 10.** Mean OmF (left column) and OmA (right column) as a function of latitude (binsize 2°) and time (binsize 1 day) for the simultaneous assimilation of GOME-2 and OMI.

basis for latitude bins with a size of 2°. In Figure 10 these column integrated OmF and OmA are shown as function of latitude and time.

The highest values of the OmF and OmA are observed at high latitudes around the polar night. The wavelength change in Band 1B is clearly visible for the GOME-2 data. Another step change in the OmA is visible for both the GOME-2 and OMI assimilation runs at the start of 2011, which coincides with an update of the bias correction parameters. OMI data is missing at latitudes larger than 80 degrees for 2009 onward due to the row anomaly.

## 6.3 Expected and observed OmF

The OmF of the results should be consistent with the uncertainties of the observations and the model forecast. The expected OmF is based on the observation error and the forecast error and is the mean of the square root term in the right hand side of equation (13) for all observations in a given layer. The observed OmF for each layer for the whole assimilation period on the



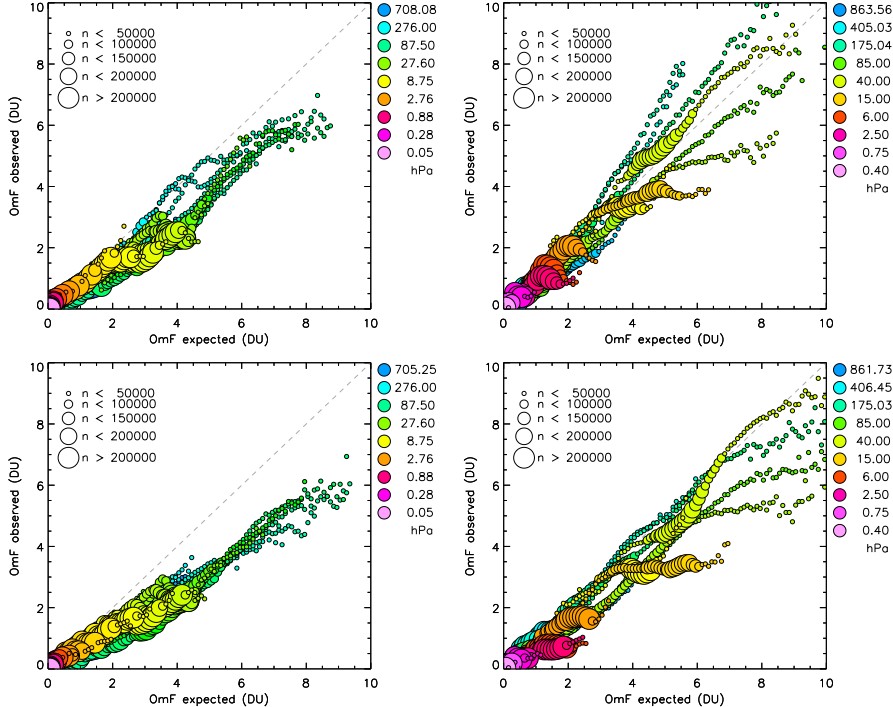

**Figure 11.** Observed vs. expected OmF. Top left: assimilation of GOME-2 only, top right: assimilation of OMI only. Bottom row: results from the simultaneous assimilation of both GOME-2 and OMI. Bottom left: GOME-2, bottom right: OMI. Colors indicate the pressure levels. Note that not all levels are plotted in the legend while all levels are plotted in the figure. The size of the circles gives the number of assimilated pixels in that respective OmF-bin (bin-size = 0.2 DU).

other hand is the mean of the left hand side of equation (13). In Figure 11, the observed OmF is plotted as a function of the expected OmF for the model runs with assimilation of GOME-2 only, with assimilation of OMI only, and for both instruments separate with the data taken from the model run with simultaneous assimilation.

Note that the pressure levels are those from the observations, not the regridded levels used in the calculation of the OmF and OmA above. The expected and observed OmF are close to the 1-to-1 line, which shows that the model error $\sigma_{x^f}$ is of the correct magnitude for the current observations. The expected and observed OmF are somewhat closer to the 1-to-1 line in the case of the simultaneous assimilation of GOME-2 and OMI than for the assimilation of each instrument independently. The model error that is used is therefore probably slightly better suited for the assimilation of multiple instruments simultaneously than for the assimilation of a single sensor.

## 6.4 Assimilation validation with sondes

The model output was validated against ozone sondes that were obtained from the World Ozone and Ultraviolet Radiation Data Center (WOUDC, WMO/GAW, 2016), see Figure 12. This is the same ozone dataset as was used to derive the bias




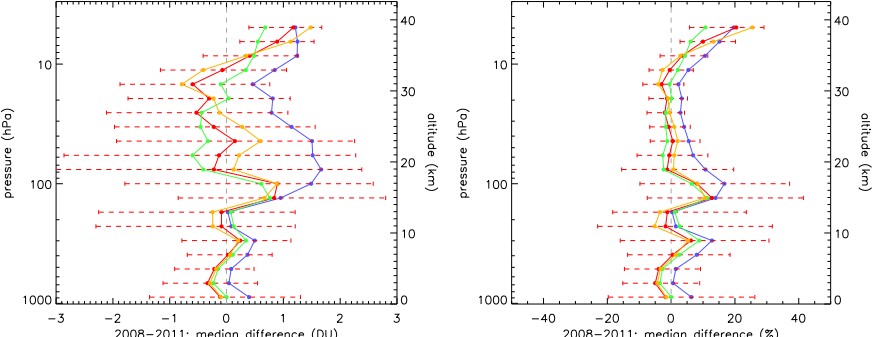

**Figure 12.** Validation of the model runs for 2008-2011. Left: the median of the absolute difference in DU, right: the median of the relative differences. Blue: model run without assimilation, green: model run with assimilation of GOME-2 only, yellow: run with assimilation of OMI only, red: assimilation of both GOME-2 and OMI. The error bars are plotted for the simultaneous assimilation only, and range from the 25% to the 75% percentile.

correction. Note however that many more observations are assimilated than were used deriving the bias correction, while all observations are corrected with the same factor. The assimilation model runs are significantly better than the free model run. This is especially true for the part of the atmosphere where GOME-2 and OMI are most sensitive to the ozone concentration, between 100 and 10 hPa. In this area, the model run with assimilation of GOME-2 only shows a negative bias with respect to

the ozone sondes, while the assimilation of OMI shows a positive bias. The assimilation of both GOME-2 and OMI shows the smallest bias.

In the troposphere the assimilation also improves, but not as much as in the stratosphere. Note that in the troposphere the chemistry scheme is different than in the stratosphere (see section 3). The assimilation shows a deviation in the tropopause, between 200 and 100 hPa, although the L2 data does not show such large biases (see Figure 5). The vertical resolution of

model and observation is different, therefore the ozone from the observation has to be redistributed over the model layers, a process which is included in the operator $H$. A small error in the redistribution of ozone in a region with a strong gradient in the concentration (such as the tropopause) will result in large uncertainties in the ozone concentration at this altitude. Above 10 hPa the assimilation shows increasing biases, and the difference with the free model run decreases. Although the L2 data also shows an increasing bias above 10 hPa, it should be noted that the number of sondes reaching this altitude is limited with

respect to the tropopause region between 200 and 100 hPa. Also, there is a representation error of the sonde with respect to the 3° longitude ×2° latitude model grid. Therefore it is not as straightforward to attribute this increase in bias to either model or observation error.





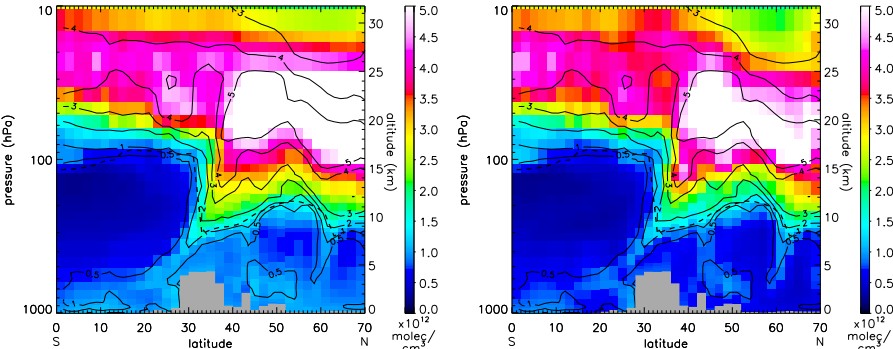

**Figure 13.** Two meridional cross sections over the Tibetan Plateau, located at $84.25°$E on 25-02-2008, 6 UTC. The colors indicate the ozone concentration from the free model run (left) and the assimilation of both GOME-2 and OMI (right). The solid contours show the ozone concentrations from the ERA-Interim reanalysis. The dashed line shows the thermal tropopause.

## 7 Case study

To demonstrate the performance of the assimilation algorithm we analysed the results for a day above the Tibetan Plateau (located between $30°$N and $40°$N), where a highly dynamical atmosphere exists. In Figure 13, ozone concentrations from the ERA-Interim reanalysis (Dee et al., 2011; Dragani, 2011) are plotted as contours over the ozone concentrations from the model runs with and without simultaneous assimilation of GOME-2 and OMI. There is a significantly better agreement between the two datasets north of $35°$N at pressure levels between 70 and 10 hPa. Even though the GOME-2 and OMI instruments have limited sensitivity in the troposphere, the tropospheric ozone concentrations of the ERA-Interim reanalysis and assimilated tropospheric ozone are in better agreement north of the Tibetan Plateau. There are also two stratosphere-troposphere exchanges (STE) visible, at $30°$N and $60°$N. These STEs are associated with strong jet-streams (perpendicular to the page) reaching wind speeds of up to 50 m/s at 250 hPa.

## 8 Discussion

When two instruments are assimilated simultaneously, their differences should be taken into account. For example, the algorithms used for the retrieval of GOME-2 and OMI ozone profiles both produce partial columns. However, the number of layers in the retrievals differ and the sensitivity of the retrieval is expressed by the averaging kernel. Both the different vertical resolution and the averaging kernel are incorporated into the observation operator $H$. Both instruments have different horizontal resolution, something which has not been taken into account in the current version of the assimilation algorithm. The measurement principle of GOME-2 (i.e. a cross-track scanning mirror) is different than that of OMI (i.e. a fixed 2D CCD detector). As a result, the ground pixel size of GOME-2 is constant, while that of OMI varies across the track. Therefore, the representation error of OMI will increase towards the edges of the swath. The effect of the changing OMI footprint size has not been investigated. A more thorough check on the instruments behaviour throughout time might have revealed the effect of





the OMI L0 to L1b processor update sooner. The threshold of the parameter in the OmF filter might be made instrument and time dependent in order to minimise the effect on the number of assimilated pixels.

Two different instruments can be biased with respect to each other. In order to minimise the bias, a bias correction scheme has been implemented with respect to ozone sondes. We used cloud free observations (max. cloud fraction 0.20) for the bias

correction in order to get a maximum amount of information from the troposphere. As a consequence, we could not use all available sondes in deriving the bias correction. Sudden changes in the bias correction parameters are visible at the start of the year, when the parameters are changed. To minimise these changes, it might be worthwhile to implement an interpolation scheme for the bias correction parameters similar as for the MSR data (see van der A et al., 2010, 2015).

The model can run a full chemistry scheme, but instead a parameterised chemistry scheme has been used in favour of speed.

Another possibility to increase the accuracy of the model is to increase the horizontal resolution from the current $3° \times 2°$ (lon $\times$ lat) to for example $1° \times 1°$. However, in both cases it might be necessary to reduce the vertical resolution of the model to keep the computational cost at an acceptable level.

The model covariance matrix is also an expensive step in the assimilation algorithm. We have reduced the calculation cost by parameterising it into a time dependent error field and a time independent correlation field. The data from April 2008 was used

to derive the correlations, which were then used for the whole assimilation period. The assumption that the derived correlations are constant throughout time has not been tested due to lack of time and resources.

## 9   Conclusions

An algorithm for the simultaneous assimilation of GOME-2 and OMI ozone profiles has been described. The algorithm uses a Kalman filter to assimilate the ozone profiles into the TM5 chemical transport model. Compared to previous versions, the

algorithm is significantly updated. The observational error has been characterised using a newly developed in-flight calibration method. Since the Kalman filter equations are too expensive to calculate directly for the current setup, the model covariance matrix is divided into a time dependent error field and a time independent correlation field. The time evolution is applied to the error field only, while the correlation is assumed to be constant. The model error growth is modelled by a new function that prevents the error from increasing indefinitely, and the correlation field has been newly derived using the NMC method. Large

biases between retrievals of the two instruments might destabilise the assimilation. To avoid this, a bias correction using global ozone sonde observations has been applied to the retrieved ozone profiles before assimilation.

Four model runs were performed spanning the years between 2008 and 2011: without assimilation, with assimilation of GOME-2 only, with assimilation of OMI profiles only and with simultaneous assimilation of both GOME-2 and OMI profiles. Depending on the altitude, the OmF and OmA for one instrument might be larger than the other, which might change in the

course of time. Assimilating the observations from these instruments simultaneously, increases the overall sensitivity of the assimilation. Two notable instrumental effects are the band 1A/1B wavelength shift for GOME-2, which causes a significant decrease in OmF and OmA. For OMI, after the L0 to L1b processor update on 1-2-2010, the uncertainty in the observations is too small with respect to the method of in-flight validation of the uncertainties presented in this paper. This caused a decrease





in the number of assimilated observations for both GOME-2 and OMI. The OmF and OmA of the simultaneous assimilation of both instruments is between the values of the single sensor assimilation. The expected and observed OmF and OmA are more similar for the combined assimilation than for the separate assimilation. Validation with sondes from the WOUDC shows that the combined assimilation performs better than the single sensor assimilation in the region between 100 and 10 hPa where

GOME-2 and OMI are most sensitive. The ozone concentrations in the troposphere are also affected by the assimilation, even though the instruments have limited sensitivity in that region. The biases of the assimilated ozone fields are smaller than those of the observations. The assimilated ozone fields are produced at regular time intervals and have no missing data. Despite the limited vertical resolution of GOME-2 and OMI, a case study of an STE over the Tibetan plateau shows that the assimilation of ozone profiles can improve the ozone distribution in a highly dynamical region.

*Competing interests.* The authors declare that they have no conflict of interest.

*Acknowledgements.* The authors acknowledge all scientists and institutes who contributed their ozone sonde data to the World Ozone and Ultraviolet Radiation Data Center (WOUDC, WMO/GAW, 2016), and the Meteorological Service of Canada for hosting this important public database. The authors would also like to thank Pepijn Veefkind for his comments in preparation of this paper. EUMETSAT is acknowledged for providing the GOME-2 L1 data and Olaf Tuinder and Robert van Versendaal for their help in the retrieval of the GOME-2 ozone profiles.

The Dutch-Finnish OMI instrument is part of the NASA EOS Aura satellite payload. The OMI ozone profiles (OMO3PR, v003) were retrieved at NASA Goddard Earth Sciences Data and Information Services Center (GES DISC) and accessed from the local storage at the Royal Netherlands Meteorological Institute (KNMI).





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
