# Peer review of "Simultaneous assimilation of ozone profiles from multiple UV-VIS satellite instruments"

_Atmospheric Chemistry and Physics, 2017_

## Referee Comment (RC1) · Anonymous Referee #2 · 9 Oct 2017

**General comment**

The manuscript describes the creation of a data set with 3D ozone concentration fields by assimilation of satellite retrieved ozone profiles in a global CTM.

As mentioned in the manuscript, ozone is regarded an "Essential Climate Variable" by the WMO, and therefore each method that provides better insight in the global distribution and temporal variability of ozone is relevant. Although assimilation of satellite retrieved ozone observations is not new, the study presented here has a number of novel aspects. These include the focus on ozone profiles rather than total columns, and the combined assimilation of profiles from two different instruments.

[Figure]

The description of the assimilation method takes most of the manuscript. This description seems complete: it briefly describes all relevant aspects such as the ozone profiles, the simulation model, and the assimilation method. The parameterization of the covariance matrices needed for the assimilation are then described more extensively, which is relevant since these dominate the result of the assimilation. The plots shows various intersections of the correlations are very illustrative.

To my opinion, the main deficiency in the presented study is that the subject of representation error is not taken into account enough. The simulation has grid cells of about 300x200 km (at the equator), which is about 2 (GOME-2) or 20 (OMI) times larger than the satellite footprints. Therefore, one would expect less variation in the grid cell concentrations since these are more smoothed compared to the satellite observations; this should explain part of the difference between model and measurements. Also the difference in footprint size is important for this study: the OMI pixels are about 10 times smaller than the GOME-2 pixels, and therefore one could expect the difference between OMI and model to be larger than between GOME-2 and model as far as related to pixel size. The issue is to some extend addressed:

- Page 5, line 27: *"The model grid cells are $3^o \times 2^o$, much larger than the satellite ground pixels and therefore no horizontal interpolation is needed"*. But a contribution to the representation error would make sense. And what does the model assume for the sub-gridcell concentration distribution, just completely mixed in the cell?

- Page 9, lines 14-15: *"These results include a representation error due to the grid cell size of the model..."*. How does error growth take representation error into account? Representation depends on the observation type (GOME-2, OMI), while there seems only a single error growth.

- Page 15, line 3: *"For these narrow-swath observations, the model is closer to the retrieved profiles"*. The feels counter intuitive, unless the observation error in narrow-swath mode is much smaller.

- Page 20, line 15-16: *"Both instruments have different horizontal resolutions, . . . "*. This issue should not be left for the discussion section only.

- Page 20, line 18-19: *"The representation error of OMI will increase towards the edges of the swath"*. But if the footprint is in better agreement with the model grid size, the representation error will be smaller. See also the comment on Page 7, line 1.

Because it is important for the result how GOME-2 and OMI observations are weighted in the assimilation, I think the representation error deserves a more extended discussion, for example as a new section 5.2 . It is not necessary to re-run the experiments, but could the authors at least give an idea of how other representation error formulations might change the results? And what would be a proper way to improve on this; could super observations help?

Finally, the case study is described very short. Why was this event chosen, is it a common test case for ozone? A bit more text would be nice, otherwise this section does not contribute much.

**Specific comments**

- Page 2, line 32-33: *"Second, it does not produce an estimate of the uncertainty . . . "*. Think this is formulated too strong. Depending on the optimization method also 4D-var could produce an estimate of the uncertainty in terms of the inverse

Hessian of the cost function. Also ensemble methods might produce an uncertainty estimate.

- Page 4, line 16. What are typical DFS values for GOME-2 and OMI here? On page 6, line 5 a value of *"about 5 to 6"* is mentioned, is that the same for both instruments?

- Page 6, line 7: Is the threshold 0.1 an absolute number? Or relative to the largest singular value?

- Page 7, line 1: *". . . the outermost pixels are neglected, because of the large are of these pixels"*. Larger pixels would actually match better with the grid cell size, so that would be an argument to neglected the pixels in center of the field-of-view. So why neglecting the outermost pixels, higher retrieval errors maybe?

- Page 9, line 3. How is the error growth applied, as factor to the std.dev. field? What is the time $t$, a time step? Then better use $\Delta t$. In the formula on line 9 I see that for $t \to \infty$ then $e(t) \to a$, which from Table 1 seems to be in a range 0.22-0.34. If the error growth is a factor I would expect a value above 1.0, so that means it is an absolute value?

- Page 9, lines 15-16: *"Therefore, all collocations that are more than $3\sigma$ from the mean are discarded"*. This looks more like an outlier test? The reason for discarding is not clear: is it to reduce the standard deviation because it also includes a contribution from the representation error? But that is not taken into at all. Please clarify.

- Page 10: Table 1 would be more clear as figure.

- Page 11, Figure 3: Couldn't this be used to parameterize the error growth?
**Interactive comment**

- Page 11, lines 8-10: How are the soundings extended to the top of the atmosphere?

- Page 11, line 19: Number of sondes, or number of sonde observations?

- Page 11, lines 11-14: The values presented here depend on the layer thickness, and do not make much sense therefore. Only relative numbers would be useful, DU/km or DU/Pa. Same holds for Figure 5.

- Page 14, Figure 6: OmF seems always positive, is it absolute bias maybe?

- Page 15, line 11: *". . . which changes its correction parameters at the start of each year"*. This would be easy to solve, as mentioned later on page 21 at line 8.

- Page 17, lines 3-6: Text mentions specific features for GOME-2, while Figure 10 shows results for combined assimilation. How do we see the specific features?

- Page 18, line 6: *". . . somewhat closer to the 1-to-1 line . . . "*. I don't really see this back in the figure.

- Page 19, line 15: Is the representation error bigger on higher altitudes? But Figure 4 suggests longer length scales at higher altitudes.

- Page 21, line 16: *". . . due to lack of time and resources"*. Although probably true, this remark makes more sense in a project report than a scientific journal; please reformulate.

**Minor corrections**

- Page 3, line 22: Start new paragraph at *"GOME-2 . . . "*.

- Page 8, line 21: Start new sentence at *"Instead we parameterise . . . "*

- Page 9, lines 11-17: symbols 'a' should be in Italic font.

- Page 13, Figure 5: names "GOME-2" and "OMI" in the title would be useful.

- Page 13, line 8: pressure does not has km as units . . .

- Page 19, Figure 12: caption should mention *"validation with ozone soundings"*.

---

## Referee Comment (RC2) · Anonymous Referee #1 · 6 Nov 2017

General Comments:

This paper presented the first effort to perform simultaneous assimilation of ozone profile products from two UV-Visible instruments (i.e., GOME-2 and OMI) using an extended Kalman filter in which the TM5 CTM is used for forecast. The assimilation methodology is described in detail: the basic approach and the critical improvements over previous assimilation including observation error characterization using a new method, model error growth and model correlation matrix and bias correction of both GOME-2 and OMI data with ozonesonde data. The temporal variation of OmF and OmA values are discussed in detail for various assimilation runs and the assimilation is validated with ozonesonde data. It was shown that the simultaneous assimilation of both GOME-2 and OMI data improve the assimilation results of a single instrument by

reducing the mean biases especially between 100-10 hpa. This scope of the paper is very suitable for publication in ACP. It is generally well written and organized. However, this study only shows the improvement in terms of mean biases and does not show the improvement in the reduction of standard deviation of the differences (or 25-75% range), which can be more convincing as both GOME-2 and OMI data are bias corrected before the assimilation. Some of the discussion and figures about the OMI L0 to L1 processor updates and the OmF and OmA statistics requires more clarification and could be improved. Overall, I recommend it to be published after addressing the following comments.

Specific Comments:

1. In abstract, it is good to add the improvement in terms of standard deviation of the differences, which I think it is a more important criterion.

2. P3, L32, Is the retrieval done at 65 km x 48 km? According to the readme document of the OMO3PRO product, it is retrieved at 13 km x 48 km although only 1 out of 5 pixels along the track is retrieved.

3. P5, in section 4, what is the physical meaning of state vector x (i.e., model ozone profiles?) and measurement vector y (i.e., satellite retrievals of ozone profiles)? This is probably not clear to readers who are not familiar with data assimilation. What does the superscript f mean as it is not defined.

4. P5, L23, H is already defined on L13, not need to repeat here.

5. P5, in the last paragraph, is the linear interpolation performed actually using the cumulative profiles of ozone column instead of model profiles of partial ozone column (DU/layer)?

6. P6, L31, instead of using a small fraction of the retrievals, have you tried to average all retrievals to the model grid? In this way, the spatial resolution matches with the model, retrieval noise is reduced, and more retrievals are utilized to maximize the

amount of retrieval information in the assimilation process.

7. P7, L26, the integration time of 1B is always 0.1875s. Do you mean that the integration time of 283-307 nm (previously part of 1A and now part of 1B) changes from 1.5s to 0.1875s? If so, please make it clear.

8. P8, in the paragraph of L10: the update of L0 to L1 processor corrected a bug in the noise calculation of the old L0 to L1 processing, reducing the noise by a factor of approximately SQRT(2) to SQRT(5) depending on the number of integrations per observation. So the noise calculation in the updated L0 to L1 processor should be correct and better. Since your estimate of noise compares quite well with the old one, do you think if the noise calculation in the updated L0 to L1 processor is wrong or is there any limitation in your approach using equation (14)? Also, the noise difference before and after the update of the L0 to L1 processor should be a factor of ∼ SQRT(2) to SQRT(5), not a factor of 5. Furthermore, all the OMI level 1b data have been reprocessed with the new processor. So for the OMI ozone profile product before February 1, 2010, has it been reprocessed using the new level 1b data? In Figure 2, left panel is for Feb 25, 2006 and the right panel is for Feb 5, 2010. I think that it is even better to compare the data from the same day, one with older processor and one with the updated processor.

9. P9, L4, what is the meaning of A in the equation?

10. Table 1 caption, it is useful to add the meaning of a (i.e., maximum relative error of the model)

11. P10, L22, why using fitted correlations rather than the calculated correlations?

12. In Section 5.4, what are the coincidence criteria (e.g., time difference and distance) between GOME-2/OMI and ozone sonde observations? What are the ozonesonde stations used in this study? It is good to add a table of ozonesonde locations and the number of profiles used at each location.

13. P11, L15-16, please make it clear what latitude bands this figure is for? Or do you

mean all the data at all latitudes?

14. P14, Why the 1A/1B boundary in GOME-2 change decreases the surface layer OmF and OmA as the wavelengths contributing to the surface layer retrievals are longer wavelengths (e.g., 2B) that do not change.

15. P15, the sudden change at the start of 2009 for GOME-2 above 10 hPa is likely due to the change of 1A/1B boundary in December 2008.

16. In section 6.1, the OmF and OmA for the assimilation of one single instrument is not mentioned and discussed in this section while the conclusion mentions that "The OmF and OmA of the simultaneous assimilation of both is between . . . (P22, L1-2)." I suggest adding OmF and/or OmA for the assimilation of one single instrument on Figures 6 and 7 and add some discussion. Also the sentence in the conclusion is not clear: the OmF and OmA values are calculated for each instrument even for simultaneous assimilation, so it is not clear about what "between" mean. Are OmF and OmA values for GOME-2/OMI of simultaneous assimilation smaller than those of single assimilation? 17. P16, L13, what do you mean "the OmF differences become so large?" Do you mean the difference between OmF for GOME-2 and OmF for OMI? Also it is not clear about "only the OMI data have changed." Please clarify it.

18. P18, L6, in "The expected and observed OmF are somewhat closer to the 1-to-1 line" and in conclusion (P22, L2-3), Figure 11 seems to contradict to this as more data points are clearly far away from 1-to-1 line in the bottom panels. Also please add slope and correlation to show the improvement quantitatively.

19. Section 6.4, because the GOME-2, OMI retrievals are bias-corrected, it is more useful to examine the standard deviations of the differences between model/assimilations and ozone sonde (or 25%/75% percentiles of the differences) than showing the median differences to demonstrate the improvement quantitatively. Please add a figure or add panels to Figure 12 to show the comparisons of these quantities for the four runs.

Technical comments 1. P1, L2, change "satellite" to "satellites"

2. P1, L16, change to "designated as"

3. P2, L8, change "((S)BUV)" to "SBUV"

4. P2, L13, move "observed" before "once or twice a day"

5. P2, L14, change it to "Ultraviolet"

6. P3, L7, change "a tropospheric ozone column" to "the tropospheric ozone column"

7. P3, L16-17, change it to "and in section 3 the chemical transport model that we use for the data assimilation is described" or "and section 3 describes the chemical transport model that we use for the data assimilation"

8. P3, L32, change to "two UV"

9. P4, L30, change to "meteorological"

10. P13, L2, change "An important diagnostic . . .. are the differences" to "An important diagnostic . . . is the difference"

11. P17, L5, change "OMI data is missing" to "OMI data are missing"

---

## Author Comment (AC1) · 20 Dec 2017

**Answers to the referee comments for article "Simultaneous assimilation of ozone profiles from multiple UV-VIS satellite instruments"**

The accompanying difference document has been generated with the latexdiff program. This program visually highlights all changes in the document, so we do not repeat all edited text in these answers. At the request of referee #2, a new figure has been created instead of table 1 so other figure numbers will have changed. These new figure numbers are not highlighted by the latexdiff program. Note that references to pages, lines, figures and tables in this "Answers..." document refer to the ACP discussion paper, not to the difference paper unless otherwise noted.

**1   Anonymous referee #1**

**1.1   Specific comments**

**1. In abstract, it is good to add the improvement in terms of standard deviation of the differences, which I think it is a more important criterion.**
Please see the answer to specific comment 19.

**2. P3, L32, Is the retrieval done at 65 km x 48 km? According to the readme document of the OMO3PRO product, it is retrieved at 13 km x 48 km although only 1 out of 5 pixels along the track is retrieved.**
The readme doc the referee is probably referring to, can be found online at `https://aura.gesdisc.eosdis.nasa.gov/data/Aura_OMI_Level2/OMO3PR.003/doc/README.OMO3PR.pdf` Indeed, one in 5 scanlines in the flight direction is skipped due to limited processing resources, thus the pixel size is $13 \times 48$ km. The text of the paper has been updated.

**3. P5, in section 4, what is the physical meaning of state vector x (i.e., model ozone profiles?) and measurement vector y (i.e., satellite retrievals of ozone profiles)? This is probably not clear to readers who are not familiar with data assimilation. What does the superscript f mean as it is not defined.**
The state vector x is the model ozone distribution and y are the retrieved ozone profiles. Superscripts $f$ and $a$ are forecast and analysis respectively. The text of the paper has been updated.

**4. P5, L23, H is already defined on L13, not need to repeat here.**
There italic $H$ (the observation operator) has been removed, but the boldface $\mathbf{H}$ (the sensitivity of the observation operator with respect to the state) is still explained in the text because they refer to different quantities.

**5. P5, in the last paragraph, is the linear interpolation performed actually using the cumulative profiles of ozone column instead of model profiles of partial ozone column (DU/layer)?**
The linear interpolation is performed on the model partial ozone column profile. The text of the paper has been updated.

**6. P6, L31, instead of using a small fraction of the retrievals, have you tried to average all retrievals to the model grid? In this way, the spatial resolution matches with the model, retrieval noise is reduced, and more retrievals are utilized to maximize the amount of retrieval information in the ssimilation process.**

Referee #2 (i.e. page C3) refers to this possibility as superobservations. The assimilation algorithm described in this paper requires the averaging kernel to account for the instrument sensitivity. The AK is a retrieval specific quantity and averaging AKs is not straightforward. The text of the paper has been updated.

**7. P7, L26, the integration time of 1B is always 0.1875s. Do you mean that the integration time of 283-307 nm (previously part of 1A and now part of 1B) changes from 1.5s to 0.1875s? If so, please make it clear.**

That is indeed what we mean and the text of the paper has been updated.

**8. P8, in the paragraph of L10: the update of L0 to L1 processor corrected a bug in the noise calculation of the old L0 to L1 processing, reducing the noise by a factor of approximately SQRT(2) to SQRT(5) depending on the number of integrations per observation. So the noise calculation in the updated L0 to L1 processor should be correct and better. Since your estimate of noise compares quite well with the old one, do you think if the noise calculation in the updated L0 to L1 processor is wrong or is there any limitation in your approach using equation (14)? Also, the noise difference before and after the update of the L0 to L1 processor should be a factor of SQRT(2) to SQRT(5), not a factor of 5. Furthermore, all the OMI level 1b data have been reprocessed with the new processor. So for the OMI ozone profile product before February 1, 2010, has it been reprocessed using the new level 1b data? In Figure 2, left panel is for Feb 25, 2006 and the right panel is for Feb 5, 2010. I think that it is even better to compare the data from the same day, one with older processor and one with the updated processor.**

We believe there is room for improvement in the noise calculation in the L0 to L1 processor because our approach works correctly for GOME-2 and the old OMI processor version. In figure 1 in the article, we demonstrate that an instrument feature such as the GOME-2 band 1A/1B shift is correctly detected by our approach. Therefore, we do not think there is a limitation in our approach.

According to the document "Bug fix for GDPS measurement noise calculation algorithm" (TN-OMIE-KNMI-935, Issue 1, 14 October 2009), the old and new error calculation are respectively:

$$Noise\,(i,j)_k = \sqrt{S_{input}\,(i,j)_k + N_{co-addition} \cdot \sigma^2_{read-out,k,g}} \qquad (1)$$

$$Noise\,(i,j)_k = \frac{\sqrt{N_{co-addition} \cdot S_{input}\,(i,j)_k + N_{co-addition} \cdot \sigma^2_{read-out,k,g}}}{N_{co-addition}} \qquad (2)$$

where $S_{input}\,(i,j)_k$ is the signal in electrons for column $i$, row $j$ and channel $k$. $N_{co-addition}$ is the number of co-additions (i.e. 2 to 5 as the referee already mentioned) and $\sigma_{read-out,k,g}$ is the read-out noise for channel $k$ and gain switch column identifier $g$. From these two equations, it follows that the noise reduction depends on the ratio of $S_{input}$ and $\sigma_{read-out}$.

It is therefore not straightforward that the reduction should be a factor of $\sqrt{2}$ to $\sqrt{5}$ as the referee remarks.

The OMI L1b error has been updated, but the radiances have not been recalculated. No reprocessing of the ozone retrievals have been done, the new noise calculation has been used after the processor update, but not before.

We do not have data for the same day processed with the old and new processor, so unfortunately it is not possible to compare the data as the referee suggests.

**9. P9, L4, what is the meaning of A in the equation?**
"A" is a fit parameter used to fit the function $A \cdot t^{1/3}$ to the data in Eskes (2003) figure 2. The text of the paper has been updated.

**10. Table 1 caption, it is useful to add the meaning of a (i.e., maximum relative error of the model)**
Referee #2 (i.e. specific comments on page C4) prefers the table as a figure. In the caption of the new figure, we added the meaning of $a$.

**11. P10, L22, why using fitted correlations rather than the calculated correlations?** Over large distances, the calculated correlations are small and dominated by noise. Therefore the correlation was fitted and small values were set to 0.

**12. In Section 5.4, what are the coincidence criteria (e.g., time difference and distance) between GOME-2/OMI and ozone sonde observations? What are the ozonesonde stations used in this study? It is good to add a table of ozonesonde locations and the number of profiles used at each location.**
The coincidence criteria and table have been added to the text.

**13. P11, L15-16, please make it clear what latitude bands this figure is for? Or do you mean all the data at all latitudes?**
We mean the data at all latitudes. The word 'global' has been added when the figure is first mentioned in the text and to the caption itself.

**14. P14, Why the 1A/1B boundary in GOME-2 change decreases the surface layer OmF and OmA as the wavelengths contributing to the surface layer retrievals are longer wavelengths (e.g., 2B) that do not change.**
In a retrieval, the results for different layers are related as given by the covariance matrix and the averaging kernel. It is therefore possible that a change in the short wavelength end of the spectrum affects the results in an altitude region where the radiation itself does not penetrate. The text of the paper has been updated.

**15. P15, the sudden change at the start of 2009 for GOME-2 above 10 hPa is likely due to the change of 1A/1B boundary in December 2008.**
If you look very carefully at the top left panel of figure 6, change in OmF and OmA as a result of the band 1A/1B change occurs really on December 10th. The sudden changes mentioned here occur on January 1st and they are also visible in the mean OmF and OmA in figure 10 for 1-1-2011. We therefore think that the sudden changes are not related to the band 1A/1B shift.The text of the paper has been updated.

**16. In section 6.1, the OmF and OmA for the assimilation of one single instrument is not mentioned and discussed in this section while the conclusion mentions that "The OmF and OmA of the simultaneous assimilation of both is between . . . (P22, L1-2)." I suggest adding OmF and/or OmA for the assimilation of one single instrument on Figures 6 and 7 and add some discussion. Also the sentence in the conclusion is not clear: the OmF and OmA values are calculated for each instrument even for simultaneous assimilation, so it is not clear about what "between" mean. Are OmF and OmA values for GOME-2/OMI of simultaneous assimilation smaller than those of single assimilation?**

We did not mention the OmF and OmA for the assimilation of one single instrument because we want to focus on the simultaneous assimilation. One of the issues we encountered when preparing the article was that the OmF and OmA of GOME-2 and OMI seemed to differ, while the model validation results were comparable. However, there are different layers in the retrieval of GOME-2 and OMI, which affects the OmF and OmA. After the the profiles had been regridded to the same pressure levels (as described on p13, l7-9) the OmF and OmA differences between the two instruments largely disappeared. The line in the conclusion refers to a figure that we created before the vertical regridding and has been removed.

**17. P16, L13, what do you mean "the OmF differences become so large?" Do you mean the difference between OmF for GOME-2 and OmF for OMI? Also it is not clear about "only the OMI data have changed." Please clarify it.**

The section referred to by the referee has been clarified and now reads "For the simultaneous assimilation, the assimilation results may be fluctuating between OMI and GOME-2 observations if a bias exists. This might result in higher assimilation errors. Therefore, the OmF filter (see section 4, equation 13) rejects observations from both GOME-2 and OMI, even though only the uncertainties from one of the instruments (i.e. OMI) have changed."

**18. P18, L6, in "The expected and observed OmF are somewhat closer to the 1-to-1 line" and in conclusion (P22, L2-3), Figure 11 seems to contradict to this as more data points are clearly far away from 1-to-1 line in the bottom panels. Also please add slope and correlation to show the improvement quantitatively.**

Somehow, the top and bottom rows were interchanged. This has been fixed, and the parameters for the best fit line and the correlation have been added. Each panel now also has a title indicating the instrument and the model run from which the data has been used.

**19. Section 6.4, because the GOME-2, OMI retrievals are bias-corrected, it is more useful to examine the standard deviations of the differences between model/assimilations and ozone sonde (or 25%/75% percentiles of the differences) than showing the median differences to demonstrate the improvement quantitatively. Please add a figure or add panels to Figure 12 to show the comparisons of these quantities for the four runs.**

The deviation in the differences (whether standard deviation or 25%-75% percentile) are very similar for the four runs (free model run, gome_2+omi, gome_2 single and omi single). That is also the reason why only the error bars for the simultaneous assimilation have

been plotted in figure 12. The deviations can be derived from figure 12, and vary between 20-55 %-points between 0 and 20 km and between 10-20 %-points between 20 and 40 km.

The mean bias after correction is zero, but there is still a deviation around it (see the red error bars in figure 5). Figure 5 also shows that the deviations of the observations are smaller after the bias correction than before. Because the deviation is already indicated in Figure 12, we decided not to add extra panels. A few lines with the values of the deviation has been added to the text.

The results of the simultaneous assimilation of GOME-2 and OMI has a smaller bias but comparable spread. This is actually a very good result since the spread of the combined observations is naturally higher than that of a single instrument.

**1.2   Technical comments**

**1. P1, L2, change "satellite" to "satellites"**
Changed
**2. P1, L16, change to "designated as"**
Changed
**3. P2, L8, change "((S)BUV)" to "SBUV"**
Changed
**4. P2, L13, move "observed" before "once or twice a day"**
Changed
**5. P2, L14, change it to "Ultraviolet"**
The capitalisation of Violet was meant to show where the first V in UV-VIS came from. Ultra Violet was changed to Ultraviolet.
**6. P3, L7, change "a tropospheric ozone column" to "the tropospheric ozone column"**
Changed
**7. P3, L16-17, change it to "and in section 3 the chemical transport model that we use for the data assimilation is described" or "and section 3 describes the chemical transport model that we use for the data assimilation"**
Changed
**8. P3, L32, change to "two UV"**
Changed
**9. P4, L30, change to "meteorological"**
Changed
**10. P13, L2, change "An important diagnostic . . .. are the differences" to "An important diagnostic . . . is the difference"**
Changed
**11. P17, L5, change "OMI data is missing" to "OMI data are missing"**
Changed

**2   Anonymous referee #2**

**2.1   General comment**

**Page 5, line 27:** *"The model grid cells are* $3° \times 2°$*, much larger than the satellite ground pixels and therefore no horizontal interpolation is needed".* **But a contribution to the representation error would make sense. And what**

does the model assume for the sub-gridcell concentration distribution, just completely mixed in the cell?

We did a small experiment, where GOME-2 and OMI data were assimilated into TM5 running on a horizontal resolution of $1° \times 1°$ (as opposed to the standard $3° \times 2°$ used in the article). The total column standard deviation of the six $1° \times 1°$ gridcells covered by a single $3° \times 2°$ gridcell is much smaller than the error on the total column. Therefore, the representation error due to the large gridcells is not significant.

During the assimilation, it is asssumed that the model ozone is completely mixed inside a gridcell. Therefore, the retrieved ozone profile is compared directly with the model profile of the gridcell containing the center coordinates of the retrieved ozone profile.

**Page 9, lines 14-15: *"These results include a representation error due to the grid cell size of the model..."*. How does error growth take representation error into account? Representation depends on the observation type (GOME-2, OMI), while there seems only a single error growth.**

The representation error in this quote refers to the point measurement made by sondes (i.e. "truth") and the larger $3° \times 2°$ (longitude $\times$ latitude) model grid cells. They do not refer to the observations made by GOME-2 and OMI. The sonde profile is compared to the average ozone concentration over the large model grid size, so the difference will be larger than for a smaller model grid size. Since that will lead to an overestimation of the error, we discarded sondes that were to far away from the mean. The resulting sondes were used to derive the value for $a$ in equation 15 and table 1.

**Page 15, line 3: *"For these narrow-swath observations, the model is closer to the retrieved profiles"*. The feels counter intuitive, unless the observation error in narrow-swath mode is much smaller.**

In narrow-swath mode, the retrieved profiles have smaller pixel footprints than in normal mode. The narrow-swath width is a factor of 6 smaller than the normal swath, so the ground pixels will be $160 \times 27$ km (along track $\times$ cross track). The pixel filtering (1 in 3 pixels for GOME-2) did not change, so more pixels will fall in the model grid cell. Therefore, the model is pulled more towards the observations and the OmF and OmA will decrease.

**Page 20, line 15-16: *"Both instruments have different horizontal resolutions, ..."*. This issue should not be left for the discussion section only.**

In section 2, it is mentioned that the size of the OMI pixels increases towards the edges of the swath. So even for a single instrument, the horizontal resolutions of the observations might be different. The different horizontal resolutions might be reflected in different uncertainties and correlations, which will affect the assimilation. We did not look specifically at this issue, which is why the second part of the sentence quoted by the referee reads "..., something that has not been taken into account in the current version of the assimilation algorithm".

**Page 20, line 18-19: *"The representation error of OMI will increase towards the edges of the swath"*. But if the footprint is in better agreement with the model grid size, the representation error will be smaller. See also the comment on Page 7, line 1.**

The representation error will be smaller with smaller pixels, because more pixels fit in the

same area and are averaged by the assimilation algorithm.

**Because it is important for the result how GOME-2 and OMI observations are weighted in the assimilation, I think the representation error deserves a more extended discussion, for example as a new section 5.2 . It is not necessary to re-run the experiments, but could the authors at least give an idea of how other representation error formulations might change the results? And what would be a proper way to improve on this; could super observations help?**

The representation error actually consists of two separate components: the first regarding the satellite observations (pixel footprint) and the model gridcells, and the second regarding the model output and ozone sondes.

To get an idea of the sub-gridcell variation of the concentration, we performed a small experiment where we assimilated the same observations (i.e. GOME-2 and OMI) into TM5 running on a $1° \times 1°$ grid (as opposed to the standard $3° \times 2°$ used in the article). The total column standard deviation of the six $1° \times 1°$ gridcells covered by a single $3° \times 2°$ gridcell is much smaller than the error on the total column. Therefore, the representation error due to the large gridcells is not significant. The text of the discussion section has been updated.

Sondes are used for deriving the model error growth (see section 5.2 and equation 15). Since sondes perform point measurements and the model output is given on large gridcells, there is a representation error. This is already discussed in section 5.2, following equation 15.

Referee #1 also inquires after the possibility of using superobservations. The assimilation algorithm described in this paper requires the averaging kernel to account for the instrument sensitivity. The AK is retrieval specific quantity and averaging AKs is not straightforward. Therefore we do not think that using superobservations could help in reducing the representation error.

**Finally, the case study is described very short. Why was this event chosen, is it a common test case for ozone? A bit more text would be nice, otherwise this section does not contribute much.**

The atmosphere over the Tibetan plateau is highly dynamic, which makes it an interesting area to study atmospheric dynamics. Because the atmosphere is so dynamic, it is also a difficult area for modelling. This case study was chosen because it is also present in the GOME-2 observations (X. Chen, J. A. Añel, Z. Su, L. de la Torre, H. Kelder, J. van Peet, and Y. Ma. The deep atmospheric boundary layer and its significance to the stratosphere and troposphere exchange over the Tibetan Plateau. PLoS ONE, 8(2):e56909, 2 2013). The observations nicely coincide with the location of the jet stream and other dynamical features. We have updated the text of section 7 to explain better why this particular case study was selected.

**2.2   Specific comments**

**Page 2, line 32-33:** *"Second, it does not produce an estimate of the uncertainty...".* **Think this is formulated too strong. Depending on the optimization method also 4D-var could produce an estimate of the uncertainty in terms of**

the inverse Hessian of the cost function. Also ensemble methods might produce an uncertainty estimate.

We meant that a 4DVAR algorithm does not directly produce an uncertainty estimate like the Kalman filter does. We agree with the referee that there are options to produce an estimate of the uncertainty, so we have changed the line in the introduction to "Second, 4DVAR does not produce a direct estimate of the uncertainty in the ozone field, although such an estimate can be derived using computationally expensive techniques."

**Page 4, line 16. What are typical DFS values for GOME-2 and OMI here? On page 6, line 5 a value of *"about 5 to 6"* is mentioned, is that the same for both instruments?**

For the cloud free retrievals over the ozone sonde stations used in this study, the mean DFS for GOME-2 is 5.0 and for OMI 5.1. The text on page 4 has been updated.

**Page 6, line 7: Is the threshold 0.1 an absolute number? Or relative to the largest singular value?**

This is an absolute value. The text has been updated.

**Page 7, line 1: *"...the outermost pixels are neglected, because of the large are of these pixels"*. Larger pixels would actually match better with the grid cell size, so that would be an argument to neglected the pixels in center of the field-of-view. So why neglecting the outermost pixels, higher retrieval errors maybe?**

These pixels are neglected due to higher retrieval errors, resulting from high solar zenith angles and viewing angles.

**Page 9, line 3. How is the error growth applied, as factor to the std.dev. field? What is the time t, a time step? Then better use $\Delta t$. In the formula on line 9 I see that for $t \to \infty$ then $e(t) \to a$, which from Table 1 seems to be in a range 0.22-0.34. If the error growth is a factor I would expect a value above 1.0, so that means it is an absolute value?**

Equation 15 gives the absolute error at time $t$ since the last assimilation of data for a gridcell. The equation is first used to calculate the current time $t$, and since in the code the model timestep is also given, the error at the end of that timestep $(t + \Delta t)$ can also be calculated.

The parameter $a$ in equation 15 is the maximum absolute error of the model at a particular altitude. In table 1, the value of $a$ is given as a relative value of the partial column because that makes it easier to compare different altitudes. The text of the article has been updated to make this distinction clear, and the table will be replaced by a figure (see "Page 10" comment below).

**Page 9, lines 15-16: *"Therefore, all collocations that are more than $3\sigma$ from the mean are discarded"*. This looks more like an outlier test? The reason for discarding is not clear: is it to reduce the standard deviation because it also includes a contribution from the representation error? But that is not taken into at all. Please clarify.**

The error growth is determined by comparing the free model run with ozone sondes. The model runs on a $3° \times 2°$ (longitude $\times$ latitude) grid, while the sondes are essentially point

sources. The difference between model and sonde will become smaller when the model resolution increases, so the current settings will overestimate the model error. In order to prevent the model error from increasing too rapidly, we decided to remove collocations that are more than $3\sigma$ from the mean. The text has been updated to clarify this issue.

**Page 10: Table 1 would be more clear as figure.**
The table has been converted into a figure. The table values are still present as strikethrough red text because of a technical latexdiff issue.

**Page 11, Figure 3: Couldn't this be used to parameterize the error growth?**
In our parameterization of the error growth, the free model run (i.e. without assimilation) is compared to ozone sondes (i.e. "the truth") to derive a value for the maximum error of the model. Figure 3 is an illustration of the NMC-method for the derivation of the correlation matrix which uses only model values. Because the model error refers to the difference with the true state of the atmosphere, we believe that the output of the NMC-method should not be used for deriving the error growth.

**Page 11, lines 8-10: How are the soundings extended to the top of the atmosphere?**
Above the burst level of the sonde, the a priori profile was used to extend the sonde profile to the top of the atmosphere. The text has been updated.

**Page 11, line 19: Number of sondes, or number of sonde observations?**
It is the number of sondes, but only 10 (GOME-2) or 33 (OMI) reached the top level. The text has been updated.

**Page 11, lines 11-14: The values presented here depend on the layer thickness, and do not make much sense therefore. Only relative numbers would be useful, DU/km or DU/Pa. Same holds for Figure 5.**
It is true that the absolute biases in DU for GOME-2 and OMI depend on the layer thickness and can therefore not be compared directly. That is why we also give the relative biases, both in the text and in figure 5, which can be compared to each other. We therefore do not see the added value of providing biases in DU/km or DU/Pa, and only added the instrument names to the plot title as requested by the referee in the "minor corrections" section below.

**Page 14, Figure 6: OmF seems always positive, is it absolute bias maybe?**
It is indeed the absolute bias. There was an error in equation 16 which now has been fixed.

**Page 15, line 11: *"...which changes its correction parameters at the start of each year"*. This would be easy to solve, as mentioned later on page 21 at line 8.**
An interpolation scheme to reduce the differences between the bias correction parameters for different years is something to be considered for a future version of the algorithm, as mentioned in the discussion.

**Page 17, lines 3-6: Text mentions specific features for GOME-2, while Figure 10 shows results for combined assimilation. How do we see the specific fea-**

tures?
The instrument specific features were visible in an earlier version of the plot which had more panels. The text has been updated.

**Page 18, line 6:** *"...somewhat closer to the 1-to-1 line...".* **I dont really see this back in the figure.**
Plot has been updated, see answer to specific comment 18 by referee #1

**Page 19, line 15: Is the representation error bigger on higher altitudes? But Figure 4 suggests longer length scales at higher altitudes.**
We try to attribute the observed increase in bias observed above 10 hPa to either model or observations. But since the observations also show an increase, it is not straightforward to do so. Because the representation error contributes to the observed bias, it has been mentioned in the text. We did not try to suggest that it increases with altitude, in fact, the representation error and the length scales are unrelated. The increase in correlation observed in Figure 4 are a consequence of the stratified structure of the stratosphere.

**Page 21, line 16:** *"...due to lack of time and resources".* **Although probably true, this remark makes more sense in a project report than a scientific journal; please reformulate.**
The last part of the sentence has been deleted.

**2.3  Minor corrections**

**Page 3, line 22: Start new paragraph at** *"GOME-2..."*
Changed
**Page 8, line 21: Start new sentence at** *"Instead we parameterise..."*
Changed
**Page 9, lines 11-17: symbols** $a$ **should be in Italic font.**
Changed
**Page 13, Figure 5: names** *"GOME-2"* **and** *"OMI"* **in the title would be useful.**
Changed
**Page 13, line 8: pressure does not has km as units...**
The levels are defined in km, converted to hPa. Changed text.
**Page 19, Figure 12: caption should mention** *"validation with ozone soundings".*

[revised manuscript text omitted]